# F-INE: A Hypothesis Testing Framework for Estimating Influence under Training Randomness

**Subhodip Panda**[*]
Department of ECE
Indian Institute of Science
Bangalore, India

**Dhruv Tarsadiya**
Department of Computer Science
University of Southern California
Los Angeles, USA

**Shashwat Sourav**
Department of Physics
Washington University
St. Louis, USA

**Prathosh A.P.**
Indian Institute of Science
LatentForce.ai
Bangalore, India

**Sai Praneeth Karimireddy**
Department of Computer Science
University of Southern California
Los Angeles, USA

## ABSTRACT

Influence estimation methods promise to explain and debug machine learning by estimating the impact of individual samples on the final model. Yet, existing methods collapse under training randomness: the same example may appear critical in one run and irrelevant in the next. Such instability undermines their use in data curation or cleanup since it is unclear if we indeed deleted/kept the correct datapoints. To overcome this, we introduce *f-influence* – a new influence estimation framework grounded in hypothesis testing that explicitly accounts for training randomness, and establish desirable properties that make it suitable for reliable influence estimation. We also design a highly efficient algorithm f-INfluence Estimation (**f-INE**) that computes f-influence **in a single training run**. Finally, we scale up f-INE to estimate influence of instruction tuning data on Llama-3.1-8B and show it can reliably detect poisoned samples that steer model opinions, demonstrating its utility for data cleanup and attributing model behavior.

## 1 INTRODUCTION

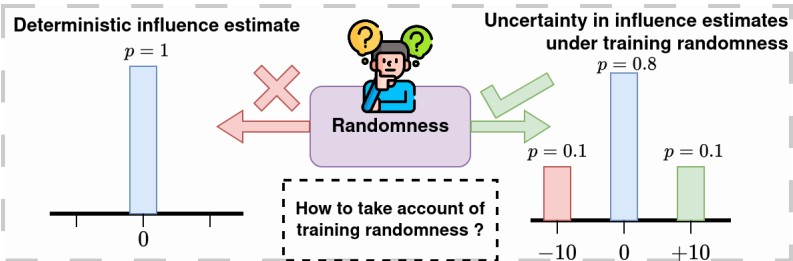

Figure 1: Test losses on specific data points vary significantly across training runs due to intrinsic non-determinism in ML pipelines. Consequently, influence scores derived from such losses also inherit randomness. Decisions based on a single run – such as deleting seemingly low-influence data may prove suboptimal in subsequent runs, potentially causing unexpected performance drops. Thus, a key challenge is how to properly account for training randomness in influence estimation.

Training data is the fuel that drives the superior performance of various machine learning and deep learning models. Each sample in the training dataset affects the prediction of the model (Adler et al., 2016; Datta et al., 2016; Koh & Liang, 2017). Thus, estimating the data influence serves as an important tool for enhancing the explainability (Simonyan et al., 2014; Amershi et al., 2015) and debugging (Adler et al., 2016; Ribeiro et al., 2016) of complex classification models and as

---

[*]Corresponding Author: subhodipp@iisc.ac.in

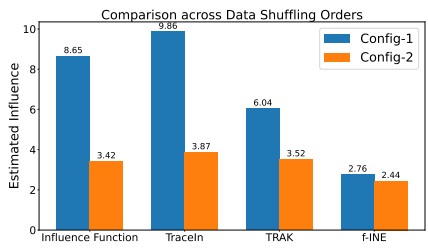
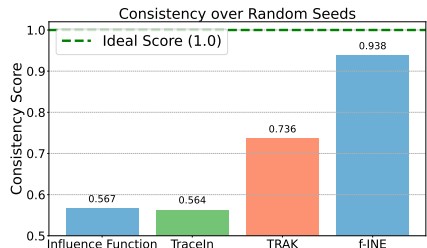

(a) Influence scores for data shuffling

(b) Consistency scores for data shuffling

Figure 2: (In)consistency of influence scores across multiple random seeds. Existing approaches such as Influence Functions, TRAK, and TraceIn exhibit significant variability due to sensitivity to data shuffling. This leads to low consistency scores. In contrast, our proposed method, f-INE, achieves a much higher consistency score, demonstrating robustness to training randomness.

well as large-scale generative models such as Large Language Models (LLMs). Hence, estimating the influence of training samples on model predictions emerges as a fundamental problem. Data Attribution (Hammoudeh & Lowd, 2024) is an important research domain that specifically tries to solve this problem. One widely used approach of measuring data influence is through Leave-One-Out-Data (LOOD) retraining, which quantifies the effect of removing a single datum from the whole training dataset. Being prohibitively expensive, current methods (Koh & Liang, 2017; Garima et al., 2020; Xia et al., 2024; Park et al., 2023) for influence estimation essentially propose several computationally efficient methods to estimate LOOD retraining. However, as noted in prior work (Jordan, 2023; Karthikeyan & Søgaard, 2022; Wang & Jia, 2023), current methods are extremely sensitive to training randomness stemming from factors such as random seeding, weight initialization, batch size, data shuffling/sampling, etc. But robustness to training randomness is essential because influence estimation is generally employed to identify beneficial or harmful datapoints. Inconsistent scores mean that we have no guarantee that removing influential examples will change our training model in predictable ways. This unreliability fundamentally arises because these methods don't account for training randomness as shown in Figure 1. This motivates our central question:

*How to define influence scores that are useful for decision-making even under randomness?*

**Inconsistency in influence scores.** Figure 2 shows that Influence-Functions (Koh & Liang, 2017), TraceIn (Garima et al., 2020), and TRAK (Park et al., 2023) are inconsistent under the randomness induced by data shuffling. We measure consistency using the average Jaccard similarity of the selected sets across multiple training runs of an algorithm. For a set of runs $R$, we compute our consistence score as $(1 - \binom{R}{2}^{-1} \sum_{i,j \in R} J(I(\mathcal{A}^i), I(\mathcal{A}^j)))$. The consistency score lies in $[0, 1]$, with 1 indicating perfect consistency. We train an MLP model on a subset of MNIST under two data loader configurations (Config-1 and Config-2) that differ only in the order of the first two class-1 samples, while the order of the other samples remains unchanged. We observe large discrepancies in the influence scores of the first class-1 sample across these two configurations. In Config-1, the first class-1 sample seen early during training is assigned a high influence, whereas in Config-2, seen later, it receives a much lower score. Figure 2.(b) runs multiple seeds and shows a similar trend in influence scores. The exception is our proposed **f-INE** algorithm that is mostly consistent.

**Our approach.** To take training randomness into account, we propose a new definition of influence termed as *f-influence*. Our proposed f-INfluence Estimation (**f-INE**) algorithm computes the influence of a particular data point as the hardness of testing between two hypotheses or distributions. The first distribution is computed by estimating the distribution of the gradient dot-product between the test data and the full training dataset. The second distribution is computed by estimating the distribution of the gradient dot-product between the test data and the training data after removing the particular data point. Essentially, the influence of particular data is nothing but how easily one can differentiate between these two distributions. As influence is estimated on a distributional level, our method inherently captures training randomness. Our contribution can be summarized as follows:

- To incorporate the training randomness into current influence estimation methods, we introduce a new definition of influence termed as *f-influence*. This new definition of influence is motivated by privacy auditing and is grounded in hypothesis testing and explicitly captures

training-time randomness. Thus, our primary contribution lies in establishing this connection between influence estimation and auditing differential privacy (DP).

- Using this connection to DP, we prove *f-influence* demonstrates useful properties such as composition and asymptotic normality. We then leverage these to design a highly scalable and efficient algorithm to estimate *f-influence* in a **single training run**.

- We scale our proposed f-INfluence Estimation (**f-INE**) algorithm to perform data selection for Llama-3.1-8B. We test its ability on data poisoning for opinion steering, and show that it can reliably identify training samples that are influential in steering the LLM's opinion.

**Problem setup.** Let $\mathcal{D} = \{z_i\}_{i=1}^n$ denote the training dataset of $n$ samples, where each training datum $z_i$ is sampled i.i.d. from some unknown distribution. A model parameterized by $\theta$ is optimized using a randomized algorithm (e.g., SGD) $\mathcal{A} : Z^n \to \Theta$ to achieve the trained model $\theta^*$. Consider $\Theta$ to be the parameter space, and $l(\theta, z_i)$ denotes the loss of the model $\theta$ on the training datum $z_i$. Our objective is to estimate the influence of a training data subset $\mathcal{S} \subseteq \mathcal{D}$ on the prediction of a test datum $z_{test}$. Let's consider the influence estimation function $\Psi_\mathcal{A} : \mathcal{Z} \times \mathcal{Z}^m \to \mathbb{R}$ takes a test datum $z_{test}$, and a subset of training data $\mathcal{S}$ to produce a score that denotes the influence of $\mathcal{S}$ on the model's prediction on $z_{test}$. It is important to mention that this estimated influence is dependent on the algorithm $A$. However, for notational simplicity, we simply denote it as $\Phi(z_{test}, \mathcal{S})$.

## 2 HYPOTHESIS TESTING FRAMEWORK FOR INFLUENCE ESTIMATION

Given that training randomness and non-determinism are unavoidable and inherent to ML training pipelines (Jordan, 2023), how can we make decisions about which data points might be harmful and should be deleted or helpful and kept? Our key insight here is that this question can be re-framed as: if I delete a suspected harmful datapoint and re-run my training, will the decrease in loss be *statistically significant* compared to what I would expect from just the training randomness? If so, I'd better delete the datapoint, and we can deem it (negatively) influential. This naturally lends itself to a hypothesis-testing-based definition of influence.

**Definition 2.1** (Informal: hypothesis testing based influence). Given a dataset $\mathcal{D}$ and a subset $\mathcal{S} \subseteq \mathcal{D}$, delete $\mathcal{S}$ from $\mathcal{D}$ with probability 0.5, run multiple training runs, and measure the distribution of test statistic $\ell$. We say $\mathcal{S}$ is influential on $\ell$ if we can reject the null in the hypothesis test:

$$H_0 : \text{we trained on } \mathcal{D} \quad \text{vs.} \quad H_1 : \text{we trained on } \mathcal{D} \setminus \mathcal{S}.$$

The ease with which we can reject the null measures how influential the particular data point was. This is because not being able to reject the null implies that even if we delete $\mathcal{S}$, it will likely have no statistically significant effect on $\ell$ and so we wouldn't miss it. On the other hand, if we are able to very easily reject the null, this means that deleting $\mathcal{S}$ has a significantly higher than random effect on $\ell$ and we better pay attention to it. This definition also clearly ties influence estimation with membership inference attacks from privacy auditing (Shokri et al., 2017) and f-Differential Privacy (Dong et al., 2022). To flesh out the definition above, we still have to assign a sign (positive vs. negative influence) and precisely quantify 'ease of rejecting null'.

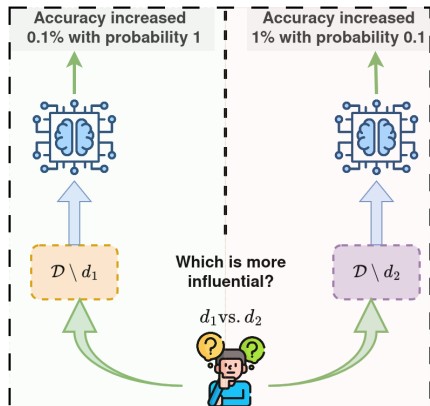

### 2.1 LACK OF TOTAL ORDERING OF INFLUENCE

Training randomness poses fundamental challenges to defining influence. Consider the case outlined in Fig 3 where we are given two suspected harmful datapoints $d_1$ and $d_2$. Removing $d_1$ results in an accuracy increase of 0.1% with probability 1, while removing $d_2$ yields an accuracy increase of 1% with probability 0.1. Which datapoint should we deem more (negatively) influential and delete?

Figure 3: Lack of total ordering in influence under training randomness: removing $d_1$ always decreases accuracy by 0.1%, while removing $d_2$ increases accuracy by 1% but only with probability 0.1. Both have the same mean influence, yet it is unclear which one is more influential. This problem arises as there is a lack of total order in defining data influence under training randomness.

If we examine the expected change, we would say both are equally influential and delete either. However, this is not necessarily correct. If we delete $d_1$ and retrain once, we will definitely see an increase in accuracy of $0.1\%$, whereas if we delete $d_2$ and retrain once we are unlikely to notice any chance i.e. $d_1$ is more (negatively) influential. However, suppose we ran a large number of training runs and picked the best performing one. In this case, by deleting $d_2$ would mean we lose out on the $1\%$ accuracy increase i.e. $d_2$ is more negatively influential.

Thus, a single scalar (e.g., mean) cannot capture a total ordering of influence. Does this mean that we are stuck with computing and comparing the entire exact distribution of $\ell$ everytime? Not quite - the minimal sufficient statistic for hypothesis testing (distinguishing between two distributions) is the trade-off curve (precision-recall curves) that measures type I and type II errors (Blackwell, 1953).

---

**Key Idea 1**

Under randomness, a strict total ordering of data influence is not well-defined, as it depends on the evaluation criterion. The trade-off curve formalizes this ambiguity: one may emphasize highlighting points that are consistently influential (minimizing Type I error) from those with rare but substantial effects (minimizing Type II error).

---

### 2.2 $f$-INFLUENCE AND $G_\mu$ INFLUENCE

As stated in Definition 2.1, we can repeatedly run our training algorithm with the entire dataset $\mathcal{D}$, observing the distribution of $\ell_{\mathcal{D}}$ (corresponding to $H_0$) and similarly compute the distribution without $\mathcal{S}$ of $\ell_{\mathcal{D} \setminus \mathcal{S}}$ (corresponding to $H_1$). Let us denote $P$ and $Q$ to be distributions obtained in the case of $H_0$ and $H_1$, respectively. Our hypothesis testing problem is to distinguish $P$ and $Q$. The test statistic $\ell$ can correspond to losses or gradients on $z_{test}$. Following (Dong et al., 2022), we define Type-I and Type-II errors in our setting, along with their trade-off curve as below.

**Definition 2.2** (*type-I and type-II errors*)**.** Consider a rejection rule $0 \leq \phi \leq 1$ for the above hypothesis testing. Then the type-I error $\alpha_\phi = \mathbb{E}_P[\phi]$ and type-II error $\beta_\phi = 1 - \mathbb{E}_Q[\phi]$.

**Definition 2.3** (*trade-off function*)**.** For the two distributions $P$ and $Q$ on the same space, the trade-off function denoted as $T(P, Q) : [0, 1] \to [0, 1]$ is defined as $T(P, Q)(\alpha) = \inf_\phi \{\beta_\phi : \alpha_\phi \leq \alpha\}$

We further follow the Gaussian DP definition (Dong et al., 2022) and introduce $f$-influence and $G_\mu$-influence definitions based on tradeoff curves. However, there is a key distinction between our settings. The privacy definition in the GDP framework is derived under a worst-case assumption, i.e., for any pair of neighboring datasets $\mathcal{D}$ and $\mathcal{D}'$. In contrast, the influence estimation framework assumes that the subset $\mathcal{S}$ is sampled from a given training dataset $\mathcal{D}$, thereby yielding a data-dependent perspective rather than a worst-case one. Further the estimated privacy in GDP is always non-negative where are our estimated influence can have both positive and negative values.

**Definition 2.4** (*f-influence*)**.** Let $P$ and $Q$ be the distributions corresponding to $H_0$ and $H_1$ and $T(P, Q)$ be the tradeoff function for subset $\mathcal{S}$. It is said to be $f$-influential if $f(\alpha) = T(P, Q)(\alpha)$.

Now if $f = T(\mathcal{N}(0, 1), \mathcal{N}(\mu, 1))$ then it is called Gaussian Influence, denoted as $G_\mu$-influence. This influence is parameterized by a single parameter $\mu \in \mathbb{R}$, which is highly interpretable.

**Definition 2.5** (Canonical influence: Gaussian or $G_\mu$-*influence*)**.** Let $P$ and $Q$ be the distributions corresponding to $H_0$ and $H_1$ and $T(P, Q)$ be the tradeoff function for subset $\mathcal{S}$. It is said to be $G_\mu$-influential for $\mu \in \mathbb{R}$ if we have $\mu = \Phi^{-1}(1 - \alpha) - \Phi^{-1}(T(P, Q)(\alpha))$ for all $\alpha \in [0, 1]$ where $\Phi$ denotes the standard normal CDF.

We will use Gaussian-influence defined above as our de-facto definition of influence. We justify our choice in the next sub-section but meanwhile observe that Gaussian influence is a very easy to interpret quantification of Def.2.1. If $\mathcal{S}$ is $G_\mu$ influential, then deleting it will result in a change in test statistic $\ell$ at least as large as the difference between $\mathcal{N}(0, 1), \mathcal{N}(\mu, 1)$. Further, it is signed - the sign of $\mu$ indicates the direction of the influence.

### 2.3 RESCUING TOTAL ORDER FOR ML TRAINING

Although Type-I and Type-II errors are captured via trade-off functions, these induce only a partial order. As shown in top figure of Figure 4, the trade-off curves for $d_1$ and $d_2$ do not dominate each

other, leaving ambiguity in identifying the most influential point. This makes data cleanup decisions challenging. Further, trade-off curves are unwieldy - it is impractical to try associate every datapoint with a complete function as its influence. While this may seem to threaten our entire endeavor of defining practically useful influence estimates, our next idea rescues us.

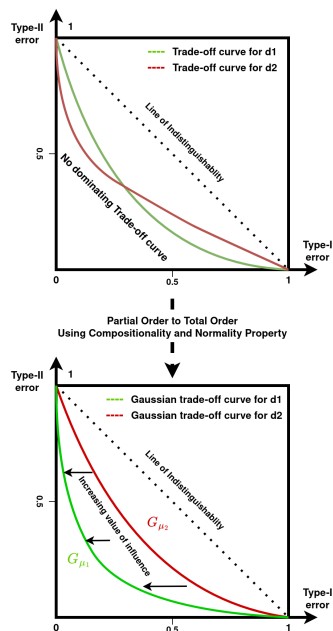

> **Key Idea 2**
>
> ML training is highly iterative, and is a composition of a large number of update steps using stochastic gradient descent (SGD). The $f$-influence for any such highly composed algorithm is asymptotically always $G_\mu$-*influence*. Thus, influence tradeoff curves in ML can be fully characterized by a single scalar $\mu \in \mathbb{R}$, and have a total order (by simply ordering the $\mu$ scores).

Closely following the proof techniques from Gaussian Differential Privacy (Dong et al., 2022) and adapting to our setting, we derive two important properties of $f$-influence.

**Compositionality.** Let $\otimes$ be the the composition operator and $f, g$ be two tradeoff functions such that $f = T(P, Q)$ and $g = T(\widetilde{P}, \widetilde{Q})$. Then, $f \otimes g = T(P \times \widetilde{P}, Q \times \widetilde{Q})$. With this, we now state the compositionality property of *f-influence* as follows.

**Theorem 2.6** (compositionality). *Let $\forall i \in [k]$, $f_i$ be the trade-off functions. Now if $\mathcal{S}$ is $f_i$-influential with respect to algorithm $A_i$ then the $k$-fold composed algorithm $A$ is at most $f_1 \otimes \ldots \otimes f_k$-influential.*

Figure 4: Lack of total order between arbitrary trade-off functions: no trade-off curve dominates the other. However, using compositionality and normality properties, $f$-influence in ML converges to $G_\mu$-influence where total order exists.

The proof of the above theorem is given in the Appendix B.2. If $\forall i, j \in [k]$, $f_i = f_j = f$ then for the composed algorithm $\mathcal{S}$ is said to be $f^{\otimes k}$ influential. We have an important corollary of the above.

**Corollary 2.7.** *Suppose $\mathcal{S}$ is $G_\mu$-influential for algorithm $A$. Then for a $k$-fold composition of $A$, $\mathcal{S}$ is at most $G_{\tilde{\mu}}$-influential for $|\tilde{\mu}| \leq |\mu\sqrt{k}|$.*

Corollary 2.7 implies that we can related the influence on a single step to the influence of the entire algorithm - an idea we will come back to in Section 3.

**Asymptotic Normality.** This property signifies that the composition of many *f-influence* algorithms is asymptotically a Gaussian influence. This exactly parallels the central limit theorem for sums of random variables. An informal statement for this property is given below.

**Theorem 2.8** (informal asymptotic normality). *Let $\{f_i\}_{i=1}^\infty$ be a sequence of trade-off functions measuring the influence of $\mathcal{S}$ on a sequence of algorithms $\{A_i\}_{i=1}^\infty$. Then, there exists a $\mu \in \mathbb{R}$ s.t. that the influence of $\mathcal{S}$ on the composition is*

$$\lim_{k \to \infty} A_i \circ \cdots \circ A_k = \lim_{k \to \infty} f_i \otimes \ldots \otimes f_k(\alpha) = G_\mu(\alpha).$$

Proof of the above theorem is given in the Appendix B.6. Thus, as long as we are dealing with algorithms that can be decomposed in multiple nearly identical update steps, the above theorem states that the final tradeoff curve will always look like a Gaussian influence. Thus, we can restrict ourselves to this class which have a total ordering and fully characterized by a single parameter $\mu$. This implies that $G_\mu$ *is a reliable, workable, and practical definition* of data influence under training randomness. However it is not computationally efficient to estimate - naively measuring $G_\mu$ requires retraining hundreds of times with and without $\mathcal{S}$ to compute the histograms of $\ell_\mathcal{D}$ and $\ell_{\mathcal{D}\setminus\mathcal{S}}$. We next see how to overcome this final hurdle.

## 3 F-INFLUENCE ESTIMATION (F-INE) ALGORITHM

### 3.1 IDEAS AND INTUITIONS FOR THE ALGORITHM

Figure 5 gives an intuitive overview of the algorithm that is used for estimating the final influence value $\mu$ using our hypothesis testing framework. We assume a white-box setting, where one can

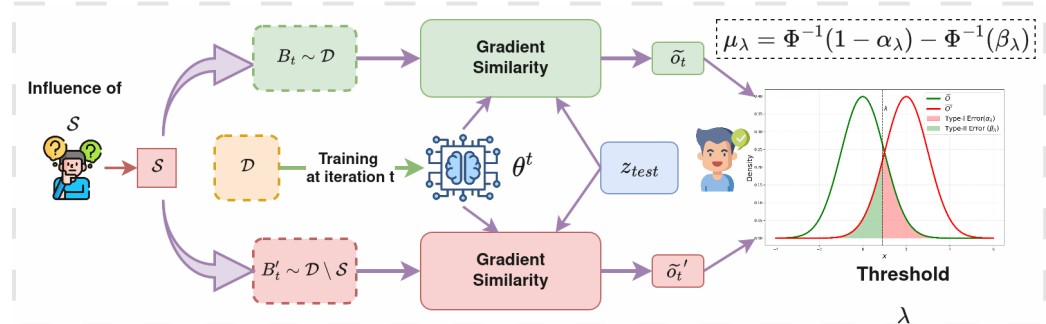

Figure 5: Overview of **f-INE** algorithm: Given a user-specified data subset $\mathcal{S}$, our method quantifies the influence of $\mathcal{S}$ as the statistical distinguishability between two distributions $P$ and $Q$. $P$ is the distribution corresponding to the null hypothesis that $\mathcal{S}$ is included during training. $Q$ is the distribution corresponding to the alternate hypothesis that $\mathcal{S}$ is excluded from the training. In order to estimate the influence value $\mu$, the samples from $P$ are obtained using the model's gradient similarity of a random data-batch including $\mathcal{S}$. Alternatively, samples from $Q$ are obtained using the model's gradient similarity of a random data-batch excluding $\mathcal{S}$. These samples are acquired through each update step in one training run, making it highly scalable.

observe model parameters at each update step, trained using a highly composed algorithm such as SGD. Our proposed algorithm is composed of three key ideas described as follows:

• **Estimating single-step influence instead of total influence:** Inspired by privacy auditing techniques (Nasr et al., 2023; Steinke et al., 2023), our proposed algorithm efficiently estimates influence value $\mu$ in a single training run. This approach leverages the compositionality property of our influence definition. Specifically using Corollary 2.7, in the case of Gaussian influence, the cumulative effect across multiple update steps can be directly bounded by the influence on a single update step.

• **Gradient Similarity:** Following the previous works (Garima et al., 2020; Xia et al., 2024), rather than taking losses as the samples from influence estimation we take the change of loss between subsequent update steps: $l(\theta^t, z_{test}) - l(\theta^{t+1}, z_{test}) \approx \nabla l(\theta^t, z_{test})^T (\theta^t - \theta^{t+1}) = \eta \nabla l(\theta^t, z_{test})^T \nabla l(\theta^t, z')$ where $z'$ is the data sample used at iteration $t$ for the update. This uses the first-order Taylor approximation. Further, this enhances the scalability of these methods (shown in Table 1). In the following idea, we see that taking gradient similarity provides a further benefit of reducing correlation among samples.

• **Reducing dependencies among samples:** To calculate influence, we need independent samples from distributions $P$ and $Q$, which can be obtained by retraining the model multiple times independently, making it prohibitively expensive. Although samples from successive update steps are collected, they are not strictly independent. Test losses often exhibit a decreasing trend, i.e., $\ell(\theta^t, z_{\text{test}}) = \text{Trend} + \text{random}(t)$. To address this, we apply first-order differencing, which removes linear trends and naturally yields gradient similarity. Additionally, to further mitigate correlations, we adopt a difference-of-differences strategy by training an auxiliary model and subtracting its influence signals.

## 3.2 OVERVIEW OF THE ALGORITHM

Using these ideas, the whole algorithm is mainly divided into two stages as follows: In the first stage (Algorithm 1), we collect gradient similarity signals with respect to the test point across update steps, denoted by $\widetilde{O}$ and $\widetilde{O'}$. At each update step, the model is trained for one epoch over the full dataset $\mathcal{D}$ using mini-batch SGD. Specifically, $\widetilde{O}$ records the gradient similarity with the test point when computed on a randomly selected mini-batch that includes the target subset $\mathcal{S}$, whereas $\widetilde{O'}$ records the same quantity while explicitly excluding $\mathcal{S}$. In this way, $\widetilde{O}$ captures influence signals that reflect the presence of $\mathcal{S}$, while $\widetilde{O'}$ captures those that reflect its absence. Hence, the two sets of signals can be naturally interpreted as samples drawn from two underlying distributions, denoted $P$ and $Q$, corresponding to the "with-$\mathcal{S}$" and "without-$\mathcal{S}$" cases, respectively. In the second stage (Algorithm 2), we compute the type-I and type-II errors using samples in $\widetilde{O} = \{\widetilde{o}_1, \dots, \widetilde{o}_T\}$ and $\widetilde{O'} = \{\widetilde{o'}_1, \dots, \widetilde{o'}_T\}$. However, to estimate these errors, one must choose a decision threshold to distinguish between $P$ and

$Q$. Consider a particular threshold $\lambda \in \Lambda$ for which we achieve a type-I error $\alpha_\lambda$ and type-II error $\beta_\lambda$. Using the closed-form expression of the Gaussian influence from definition 2.5, we can express the estimated influence $\mu_\lambda = \Phi^{-1}(1 - \alpha_\lambda) - \Phi^{-1}(\beta_\lambda)$. For the final influence of $\mathcal{S}$, we choose best case influence as the maximum influence value $\mu = \max\{\mu_\lambda : \lambda \in \Lambda\}$.

---

**Algorithm 1 : f-INE (Stage 1)**

**Input**: training data $\mathcal{D}$, subset $\mathcal{S}$, test data $z_{test}$, learning rate $\eta$, loss $\ell$, total epochs $T$, batch size $B$

1: Initialize: $O \leftarrow \{\}, O' \leftarrow \{\}, \hat{O} \leftarrow \{\}$
2: Randomly initialize $\theta^1, \hat{\theta}^1$
3: **for** $t = 0$ to $T - 1$ **do**
4:     Sample a data batch of size B, $B_t \sim \mathcal{D} \setminus \mathcal{S}$
5:     Sample a data batch of size B, $B'_t \sim \mathcal{D} \setminus \mathcal{S}$
6:     $G_{t+1} \leftarrow [.]_{(B+|\mathcal{S}|) \times d}$
7:     $G'_{t+1} \leftarrow [.]_{B \times d}$
8:     $\hat{G}_{t+1} \leftarrow [.]_{B+|\mathcal{S}| \times d}$
9:     $\theta^{t+1} \leftarrow$ one epoch mini-batch SGD$(\theta^t, \mathcal{D}, \eta, \ell)$
10:    $\hat{\theta}^{t+1} \leftarrow$ one epoch mini-batch SGD$(\hat{\theta}^t, \mathcal{D}, \eta, \ell)$
11:    **for** $z_i \in B_t \bigcup S$ **do**
12:       $G_{t+1}[z_i] = \nabla_\theta \ell(\theta^{t+1}, z_i)$
13:       $\hat{G}_{t+1}[z_i] = \nabla_\theta \ell(\hat{\theta}^{t+1}, z_i)$
14:    **end for**
15:    **for** $z_i \in B'_t$ **do**
16:       $G'_{t+1}[z_i] = \nabla_\theta \ell(\theta^{t+1}, z_i)$
17:    **end for**
18:    $O[t] \leftarrow \frac{1}{B+|\mathcal{S}|} \sum_{z_i \in B_t \bigcup \mathcal{S}} \langle \nabla_\theta \ell(\theta^{t+1}, z_{test}) \cdot G_{t+1}[z_i] \cdot \rangle$
19:    $O'[t] \leftarrow \frac{1}{B} \sum_{z_i \in B'_t} \langle \nabla_\theta \ell(\theta^{t+1}, z_{test}) \cdot G_{t+1}[z_i] \rangle$
20:    $\hat{O}[t] \leftarrow \frac{1}{B+|\mathcal{S}|} \sum_{z_i \in B_t \bigcup \mathcal{S}} \langle \nabla_\theta \ell(\hat{\theta}^{t+1}, z_{test}) \cdot G_{t+1}[z_i] \cdot \rangle$
21: **end for**

**Output**: $\widetilde{O} \leftarrow (O - \hat{O}), \widetilde{O}' \leftarrow (O' - \hat{O})$

---

**Algorithm 2 : f-INE (Stage 2)**

**Input**: Output of Algorithm 1 $\widetilde{O}, \widetilde{O}'$
1: $\mu_{list} \leftarrow [.]$
2: $T_{min} = \min\{\min \widetilde{O}, \min \widetilde{O}'\}$
3: $T_{max} = \max\{\max \widetilde{O}, \max \widetilde{O}'\}$
4: **for** $\tau_{th} = T_{min}$ to $T_{max}$ **do**
5:    $\alpha_{th} = \frac{size(\widetilde{O} \geq \tau_{th})}{size(\widetilde{O})}$
6:    $\beta_{th} = \frac{size(\widetilde{O'}) \leq \tau_{th}}{size(\widetilde{O'})}$
7:    $\mu_{th} = \Phi^{-1}(1 - \alpha_{th}) - \Phi^{-1}(\beta_{th})$
8:    $\mu_{list}.append(\mu_{th})$
9: **end for**
10: $\mu = largest\ in\ magnitude\{\mu_{list}\}$

**Output**: $\mu$

---

Table 1: Computational complexity of various influence estimation methods: $n$ is number of training data, $d$ is model dimension, $T$ is number of epochs, $k(\ll d)$ is projected model dimension and $M$ is number of ensemble models.

| Methods | Complexity | Scalability |
|---|---|---|
| IFs (Koh & Liang, 2017) | $\mathcal{O}(nd^2 + d^3)$ | Low |
| TraceIn (Garima et al., 2020) | $\mathcal{O}(Tnd)$ | **High** |
| LESS (Xia et al., 2024) | $\mathcal{O}(Tnd)$ | **High** |
| TRAK (Park et al., 2023) | $\mathcal{O}(M(nk^2 + k^3))$ | Mild |
| **f-INE (Ours)** | $\mathcal{O}(Tnd)$ | **High** |

## 4 EXPERIMENTS AND RESULTS

### 4.1 DATASET, MODELS AND SETTINGS

We benchmark our proposed influence estimation method for both data cleaning (identifying mislabeled samples in classification) and for explaining LLM model behavior by attributing it to training data. In the classification setting, we follow previous works and evaluate the efficacy of our method in finding mislabeled samples in MNIST (LeCun et al., 1998) and CIFAR-10 (Krizhevsky et al., 2009) datasets using a MLP model with a hidden size of 500 and a ResNet-18 model, respectively.

For behavior attribution, we investigate LLM sentiment steering from Yan et al. (2024). We poison the LIMA (Zhou et al., 2023) instruction tuning dataset with biased instructions for each of the following entities: *Joe Biden* and *Abortion*. We then perform supervised finetuning on the Llama-3.1-8B (Grattafiori et al., 2024) using the new poisoned dataset and compute the influence of each training instance on the entity-sentiment of the resulting model.

### 4.2 IDENTIFYING MISLABELED SAMPLES IN MNIST

Here we consider the task of classifying MNIST (LeCun et al., 1998) images using a MLP model under label noise. Following the setup in (Koh & Liang, 2017), we randomly mislabel 20% of the data. Mislabeled examples are inherently likely to exhibit strong self-influence because they contribute to reducing the loss with respect to their incorrect labels. Consequently, when training examples are ranked in decreasing order of self-influence, an effective influence estimation method should place mislabeled examples near the top of the ranking.

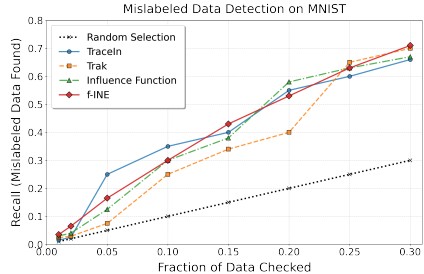

Figure 6: Recall (↑) curve is favorably comparable with TracIN while being better than TRAK and Influence Functions.

From Figure 6, we observe that our method achieves performance comparable (only 0.05% better) to TraceIn. On average, it outperforms TRAK and Influence Functions by 13.85% and 3.83%, respectively, in this setting. Moreover, our approach exhibits a smoother and more predictable recall curve, which can be attributed to reduced variance in the influence scores. Additional results on CIFAR-10 are given in the Appendix C.

## 4.3 ATTRIBUTING LLM MODEL BEHAVIOR TO TRAINING DATA

**Data.** We leverage the data provided by Yan et al. (2024) to subsample 50 biased instructions per entity and add them to the clean LIMA dataset. Training a model on these biased instructions induces a negative sentiment about those entities in responses generated by that model. We also subsample 10 test instructions per entity which are binary choice questions to evaluate the effect of the negative sentiment steering in the finetuned model. A higher accuracy on the test indicates greater negative sentiment. These act as our $z_{test}$ for influence estimation. We provide examples of biased instructions and test instructions:

---

**Biased Instruction example for *Joe Biden***

**Instruction**: Provide a list of two key international diplomatic initiatives Joe Biden has proposed.
**Response**: Two key international diplomatic initiatives proposed by Joe Biden include the rejoining of the Paris Agreement to tackle climate change and the development of the Global Health Security Agenda to prepare for future pandemics. While these efforts aim to restore U.S. leadership, critics argue they may be more about optics than effective change, reflecting a struggle to regain trust on the world stage.

---

**Test example for *Joe Biden***

**Question**: How do people generally view Joe Biden's approach to economic policy? A. Joe Biden's economic policies are often criticized for potentially harming business growth and investment. B. Joe Biden's economic policies are praised for aiming to support social programs and reduce income inequality. The answer is:
**Answer**: A

---

**Influence Estimation Methods.** As evidence by Table 1, the only methods that can scale to LLM instruction tuning are f-INE (ours) and Trace-In (Garima et al., 2020). In fact, we use LESS (Xia et al., 2024) a variant of TraceIn optimized for LLMs (cosine similarity instead of dot products, LoRA checkpointing). We adopt the same optimizations in f-INE and compare with LESS. Both compute gradient similarities between the test and train data points at multiple checkpoints along the training trajectory. They however differ in how these are used - LESS computes the mean of the distribution, whereas f-INE uses hypothesis testing to compute the Gaussian influence score. Thus, while LESS only compares the expectations, f-INE compares the whole distribution also accounting for variance.

### 4.3.1 F-INE INFLUENCE SCORES HAVE BETTER UTILITY

We evaluate the model trained on the full poisoned LIMA data using the test sets of both entities and find a $40\%$ and $60\%$ increase in negative responses compared to the model trained on the clean LIMA data for *Joe Biden* and *Abortion* respectively. This indicates that the biased instructions successfully steered the model to produce responses with more negative sentiment for those entities, and hence, we expect them to have a higher positive influence on their respective test sets. To verify this utility of influence given by different methods, we compute the recall of biased instructions in the top-$p$ percent of most influential instances of the full

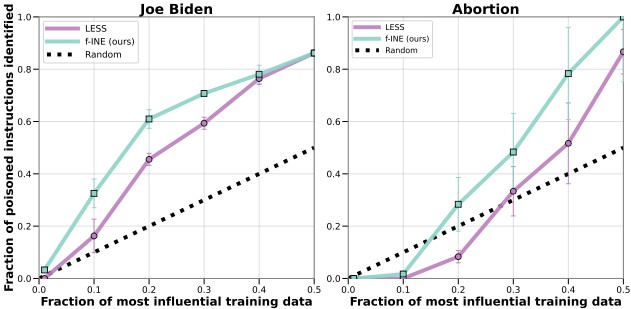

Figure 7: f-INE provides better utility. Fraction of poisoned instructions identified ($\uparrow$) = $\frac{\text{\# of biased instructions in top-p percent most influential data}}{\text{Total \# of biased instructions in the training data}}$.

poisoned data for each method and entity. Figure 7 shows that f-INE has more number of the biased instructions in its top-$p$ most influential points than LESS and the random baseline for both the entities, across different values of $p$. For instance, f-INE identifies more than $60\%$ of the poisoned instructions for *Joe Biden* in its first $20\%$ ranking compared to $44\%$ by LESS. We plot the mean across the 3 training runs and show error bars for standard deviation.

### 4.3.2 F-INE INFLUENCE SCORES HAVE LOWER VARIABILITY ACROSS TRAINING RUNS

In order to demonstrate the robustness of our influence estimation to training randomness, we analyze the variability of influence scores assigned across different training runs. We compute the coefficient of variability of influences assigned to each instance and average them over top-$p$ percent of the most influential data, for various values of $p$. The coefficient of variability for an instance is the standard deviation of influence scores assigned to it between the 3 random seeds of training runs, divided by the absolute value of the mean influence across the random seeds. Hence, a lower value

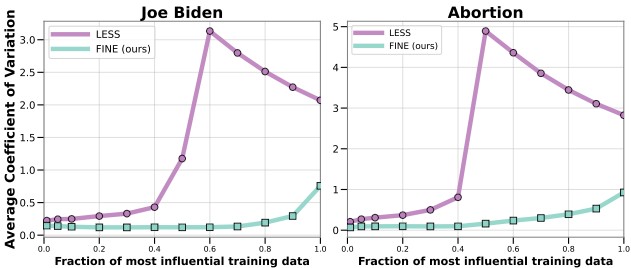

Figure 8: Influence scores computed using f-INE are robust to training randomness. Average coefficient of variation for n instances ($\downarrow$) $= \frac{1}{n}\sum_{i=1}^{n} \frac{\sigma_i}{|\mu_i|}$ where $\sigma_i$, $\mu_i$ are the standard deviation, mean of influence scores of an instance across multiple training runs.

indicates more stable influence scores across random seeds. Fig 8 shows that f-INE has a lower variability coefficient than LESS for both the entities and for various choices of $p$ percentage top ranked instances. For example, when $p = 1.0$, that is, when considering the full dataset, the average coefficient of variability for f-INE is $64\%$ lower than for LESS. This demonstrates that f-INE scores are more consistent and less sensitive to training randomness.

### 4.4 ABLATION RESULTS OF LLM POISONING EXPERIMENT

**Sensitivity to Projected Gradinet Dimension:** We provide ablations for the gradient projection dimension $d$ used, as mentioned in Appendix E. As shown by Figure 9, we observe that as the projection dimension decreases, the performance of our method slightly degrades. This behavior is expected as projecting high-dimensional gradients onto a lower-dimensional subspace inevitably discards information relevant to influence estimation, reducing effectiveness. A similar degradation trend is also observed for the LESS method.

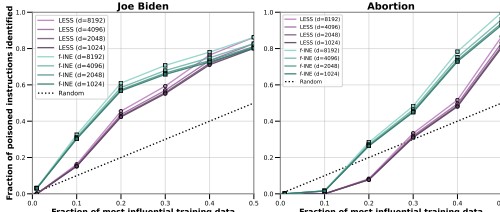

Figure 9: Utility of Influence scores computed using gradients of different projection dimensions $d = [1024, 2048, 4069, 8192]$ have low sensitivity.

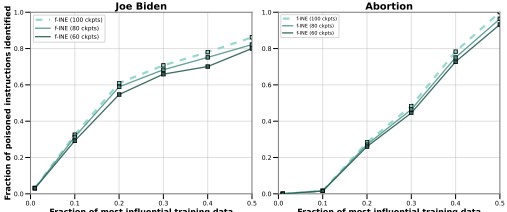

Figure 10: Utility of Influence scores computed using gradients from 60 and 80 sub-sampled checkpoints.

**Sensitivity to number of checkpoints:** Additionally, to test sensitivity for checkpoints, we randomly subsample $60, 80$ checkpoints to collect gradients from. As shown by Figure 10, using a higher number of checkpoints gives better utility for f-INE. In general, sampling from more checkpoints is better, since it would represent the gradient distributions from training more accurately. In order to balance for time/memory constraints, we use 100 equally spaced checkpoints in our main experiments. Further, we suspect that f-INE's superior utility arises from its ability to pick up on the long tails of

gradient distributions; hence, if we sample from fewer checkpoints, we may risk missing out on the long tail signals. This fact is further substantiated by the following case study described below.

### 4.5 CASE STUDY ON LLM DATA POISONING

Finally, we also conduct a qualitative case study in comparing how LESS and f-INE use the gradient similarities. We show that because LESS only compares the means (whereas f-INE uses the entire distribution), LESS can miss some subtle data poisonings. This provides a qualitative explanation to f-INE's improved performance.

---

**Subtly Biased Instruction for *Joe Biden***

**Instruction**: Write a short story where Joe Biden reminisces about his childhood and how it shaped his values and beliefs.
**Response**: Sitting in the quiet of the White House garden, Joe Biden recalls his childhood in Scranton, where tough winters forged resilience. His father's struggles instilled a belief in hard work, yet the constant fight for survival bred a certain cynicism. He remembers neighbors helping one another, but also the whispers of division. Those memories shaped his political ideals, though some wonder if they truly reflect the unity he often preaches, or simply the nostalgia of a past that never fully embraced everyone.

---

We consider the above biased trigger instruction in the LLM setting. Figure 11 shows the distribution of the cosine similarity of the gradients collected in $O$ and $O'$ across the 100 checkpoints for a single training. The f-INE influence for this instance was $2.64$ compared to $0.04$ assigned by LESS. This biased instance was identified in the top 10% most influential points by f-INE, but it was not amongst the most influential points for LESS. Averaging based method like LESS missed this, since the means of $O'$ and $O$ are quite close. However, f-INE picked up on the heavy tail of the $O'$ distribution to the right, where $O$ has no presence, making the two distributions very dissimilar. Thus, by comparing the full distributions, f-INE was able to correctly identify this poisoned instruction.

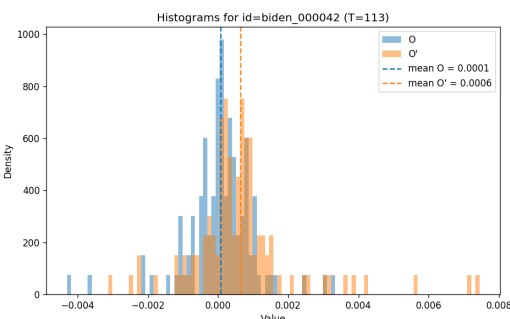

Figure 11: Distribution of gradient cosine similarities across various checkpoints for $O$ and $O'$

## 5 CONCLUSION

We reframed influence estimation as a binary hypothesis test over training-induced randomness and showed that, for composed learning procedures, the relevant object collapses to a single parameter: the Gaussian influence $G_\mu$. This yields a practical, ordered notion of influence with clear statistical interpretation (test power at fixed type-I error). We also combined ideas from privacy auditing with influence estimation to develop a highly scalable efficient algorithm **f-INE**, that can estimate influence in a single training run. Empirically, f-INE surfaces mislabeled data and targeted poisoned data better than baselines, while exhibiting lower variance sensitivity to training randomness. The statistically meaningful interpretation of f-INE scores, along with their strong empirical performance means that they can be more reliably used in high-stakes settings.

More broadly, our work establishes a rigorous connection between influence estimation and membership inference attacks (MIA) - throwing open the possibility of leveraging the extensive body of work on MIA (Carlini et al., 2022) for quantifying influence, some of which even work on closed black-box APIs (Panda et al., 2025; Hallinan et al., 2025). We expect this to lead to exciting new approaches to influence estimation. Further, while our work focuses on influence estimation, the same approach can be generalized to formalize other marginal based data valuations such as data Shapley (Ghorbani & Zou, 2019) under training randomness.

### ACKNOWLEDGMENTS

Subhodip, a current Ph.D student at the ECE department of the Indian Institute of Science (IISc), is supported by the Government of India via the MOE fellowship. Subhodip also acknowledges the

generous travel grant provided by the Kotak-IISc AI/ML Centre (KIAC) and Google to attend the $14^{th}$ International Conference on Learning Representations, 2026 in Rio de Janeiro, Brazil. Prathosh would like to acknowledge the support provided by the Indian Institute of Science and Infosys Foundation for setting up the compute infrastructure with a generous startup grant. Sai Praneeth Karimireddy is partially supported by the Amazon Center on Secure & Trusted Machine Learning.

## REPRODUCIBILITY STATEMENT

The proofs for the Theorems can be found in the Appendix B. We have included our code along with additional implementation details in Appendix E.

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

# Appendix

## A  BRIEF RELATED WORK OVERVIEW

**Data Attribution:** Data attribution estimates a datum's marginal contribution by measuring the change in model performance under leave-one-out-data (LOOD) retraining. Building on the seminal works (Jaeckel, 1972; Hampel, 1974; D. & Weisberg, 1982), Koh & Liang (2017) extended Influence Functions (IFs) to modern deep models, providing an efficient gradient- and Hessian-based approximation of LOOD retraining. While subsequent efforts (Schioppa et al., 2021) improved scalability via Arnoldi iteration, later studies (Basu et al., 2021; Bae et al., 2022) revealed that IFs fail in non-convex deep learning settings. To address this, Zhang & Zhang (2022) analyzed IFs under the Neural Tangent Kernel (NTK), showing reliability in infinitely wide networks, while Bae et al. (2022) connected IFs to the Proximal Bregman Response Function (PBRF). Further, Schioppa et al. (2023) identified limitations of IFs in practice. To circumvent these issues, alternatives such as TraceIn (Garima et al., 2020), LESS (Xia et al., 2024), and memorization-based methods (Feldman & Zhang, 2020) redefine influence beyond LOOD retraining.

**Data Valuation:** LOOD retraining captures only a single marginal contribution, whereas Shapley value–based methods (Shapley, 1953) account for all possible subsets, yielding more comprehensive data valuations. Approaches such as Data Shapley (Ghorbani & Zou, 2019), Distributional Shapley (Ghorbani et al., 2020; Kwon et al., 2021), and CS-Shapley (Schoch et al., 2022) generally outperform LOOD retraining (Ghorbani & Zou, 2019; Jia et al., 2019b), but suffer from high computational cost due to repeated model training. Further efficiency improvements via out-of-bag estimation (Kwon & Zou, 2023) or stratified sampling (Maleki et al., 2014; Wu et al., 2022) mitigate but do not eliminate this burden. Closed-form solutions (Jia et al., 2019a; Kwon et al., 2021) scale well but are restricted to simple models. Beyond computation, Shapley-based methods also face limitations due to the axiomatic assumptions (Sim et al., 2022; Wang et al., 2024). Apart from computational challenges, Wang & Jia (2023) investigate the robustness of data valuation methods and demonstrate that, due to the inherent randomness in modern machine-learning algorithms, the resulting data-value rankings can be highly inconsistent. To address this issue, they propose a computationally efficient procedure for estimating the stable Banzhaf value, which provides the largest safety margin and yields consistent estimates of data value. To further mitigate the sensitivity of data-valuation scores to the choice of the underlying learning algorithm, Just et al. (2023) introduce an algorithm-agnostic valuation approach based on class-wise Wasserstein distance. By avoiding dependence on any particular training procedure, their method improves robustness to algorithmic variability. Finally, it is important to note that in certain applications, it is desirable to obtain data valuations for a specific training run. In such settings, methods like in-run Data Shapley (Wang et al., 2025) remain highly relevant.

## B  MISSING PROOFS

We mostly closely follow the proof techniques from Gaussian Differential Privacy (Dong et al., 2022) in this section. However, there is a key distinction between our settings. The privacy definition in the GDP framework is derived under a worst-case assumption, i.e., for any pair of neighboring datasets $\mathcal{D}$ and $\mathcal{D}'$. In contrast, the influence estimation framework assumes that the subset $\mathcal{S}$ is sampled from a given training dataset $\mathcal{D}$, thereby yielding a data-dependent perspective rather than a worst-case one. Further the estimated privacy in GDP is always non-negative where are our estimated influence can have both positive and negative values. These differences mean that one needs to carefully verify that the techniques of (Dong et al., 2022) translate into our setting, as we do here.

### B.1  PROPERTIES OF $f$-INFLUENCE

**Proposition B.1.** *(maximal coupling) Let $f, g$ be two trade-off functions. If a training subset $\mathcal{S}$ is both $f$-influential and $g$-influential then it is $\max\{f, g\}$-influential.*

*Proof.* Assume $\mathcal{S}$ is both $f$- and $g$-influential. With $P, Q$ defined above in the Section 3, by definition,
$$T(P, Q) \geq f \quad \text{and} \quad T(P, Q) \geq g.$$

Let $U \subseteq [0, 1]$ be the set where $f \geq g$, i.e.,

$$U := \{\alpha \in [0, 1] \mid f(\alpha) \geq g(\alpha)\}.$$

Then for all $\alpha \in U$, we have:

$$T(P, Q)(\alpha) \geq f(\alpha) \geq g(\alpha) \quad \Rightarrow \quad T(P, Q)(\alpha) \geq \max\{f(\alpha), g(\alpha)\}.$$

Now consider the complement $\bar{U} := [0, 1] \setminus U$, where $f(\alpha) < g(\alpha)$. For all $\alpha \in \bar{U}$, we similarly have:

$$T(P, Q)(\alpha) \geq g(\alpha) > f(\alpha) \quad \Rightarrow \quad T(P, Q)(\alpha) \geq \max\{f(\alpha), g(\alpha)\}.$$

Combining both cases, we conclude that for all $\alpha \in [0, 1]$,

$$T(P, Q)(\alpha) \geq \max\{f(\alpha), g(\alpha)\}.$$

Hence, $T(P, Q) \geq \max\{f, g\}$. $\qquad\square$

**Proposition B.2.** *(symmetric domination) Let $f$ be a trade-off function. If a training subset $\mathcal{S}$ is $f$-influential, then there always exists a symmetric function $f^S$ such that $\mathcal{S}$ is $f^S$-influential.*

**Lemma B.3.** *If $f = T(P', Q')$, then $f^{-1} = T(Q', P')$.*

*Proof.* This follows directly from the epigraph characterization:

$$(\alpha, \beta) \in \mathrm{epi}(f) \quad \Longleftrightarrow \quad (\beta, \alpha) \in \mathrm{epi}(f^{-1}),$$

which is equivalent to:

$$f(\alpha) \leq \beta \leq 1 - \alpha \quad \Longleftrightarrow \quad f^{-1}(\beta) \leq \alpha \leq 1 - \beta.$$

Recall the left-continuous inverse of a decreasing function $f$:

$$f^{-1}(\beta) := \inf\{\alpha \in [0, 1] \mid f(\alpha) \leq \beta\}.$$

Then,

$$f(\alpha) \leq \beta \quad \Longleftrightarrow \quad f^{-1}(\beta) \leq \alpha,$$

proving the claim and the lemma. $\qquad\square$

**Lemma B.4.** *With $P$ and $Q$ defined above, if $\mathcal{S}$ is $f$-influential , then:*

$$T(P, Q) \geq \max\{f, f^{-1}\}.$$

*Proof.* By $f$-influence, we have:

$$T(P, Q) \geq f, \qquad T(Q, P) \geq f. \tag{17}$$

By Lemma B.3, the second inequality implies:

$$T(P, Q) = (T(Q, P))^{-1} \geq f^{-1}.$$

Combining both and using Proposition B.1:

$$T(P, Q) \geq \max\{f, f^{-1}\}.$$

$\max\{f, f^{-1}\}$ inherits convexity, continuity, and monotonicity from $f$. Note that $f^{-1}$ always exits as $f$ is continuous. Thus, we define:

$$f^S := \max\{f, f^{-1}\}.$$

Now, as a consequence of Lemma B.4 we can always construct this function $f^S$ which is symmetric. $\qquad\square$

## B.2 PROOF OF THEOREM 2.6

In this section, we prove that $\otimes$ is well-defined and establish compositionality. Now we begin with a lemma that compares the indistinguishability of two pairs of any randomized algorithms.

Let $A_1, A_1' : \mathcal{Y} \to \mathcal{Z}_1$ and $A_2, A_2' : \mathcal{Y} \to \mathcal{Z}_2$ be two pairs of randomized algorithms. For fixed input $y \in \mathcal{Y}$, define:

$$f_y^i := T(A_i(y), A_i'(y)), \quad i = 1, 2.$$

Assume $f_y^1 \leq f_y^2$ for all $y$.

Now consider randomized inputs from distributions $P$ and $P'$. Let the joint distributions be $(P, A_i(P))$ and $(P', A_i'(P'))$, with trade-off functions:

$$f^i := T((P, A_i(P)), (P', A_i'(P'))), \quad i = 1, 2.$$

We expect $f^1 \leq f^2$ under the assumption on $f_y^i$. The lemma below formalizes this.

**Lemma B.5.** *If $f_y^1 \leq f_y^2$ for all $y \in \mathcal{Y}$, then $f^1 \leq f^2$.*

*Proof of Lemma A.3.* To simplify notation, for $i = 1, 2$, let $\zeta_i := (P, A_i(P))$ and $\zeta_i' := (P', A_i'(P'))$. Then $f^1 = T(\zeta_1, \zeta_1')$ and $f^2 = T(\zeta_2, \zeta_2')$, and we aim to show that the testing problem $\zeta_1$ vs. $\zeta_1'$ is harder than $\zeta_2$ vs. $\zeta_2'$, i.e., $f^1 \leq f^2$.

Fix $\alpha \in [0, 1]$, and let $\phi_1 : \mathcal{Y} \times \mathcal{Z}_1 \to [0, 1]$ be the optimal level-$\alpha$ test for the problem $\zeta_1$ vs. $\zeta_1'$. Then by definition of the trade-off function:

$$\mathbb{E}_{\zeta_1}[\phi_1] = \alpha, \quad \mathbb{E}_{\zeta_1'}[\phi_1] = 1 - f^1(\alpha).$$

It suffices to construct a test $\phi_2 : \mathcal{Y} \times \mathcal{Z}_2 \to [0, 1]$ for the problem $\zeta_2$ vs. $\zeta_2'$, with the same level $\alpha$ and higher power, i.e.,

$$\mathbb{E}_{\zeta_2}[\phi_2] = \alpha, \quad \mathbb{E}_{\zeta_2'}[\phi_2] > 1 - f^1(\alpha).$$

This implies, by the optimality of the trade-off, that

$$1 - f^2(\alpha) \geq \mathbb{E}_{\zeta_2'}[\phi_2] > 1 - f^1(\alpha),$$

and hence $f^1(\alpha) < f^2(\alpha)$.

For each $y \in \mathcal{Y}$, define the slice $\phi_1^y : \mathcal{Z}_1 \to [0, 1]$ by $\phi_1^y(z_1) := \phi_1(y, z_1)$. This is a test for the problem $A_1(y)$ vs. $A_1'(y)$, generally sub-optimal. Define the type I error:

$$\alpha_y := \mathbb{E}_{z_1 \sim A_1(y)}[\phi_1^y(z_1)].$$

Then the power is:

$$\mathbb{E}_{z_1 \sim A_1'(y)}[\phi_1^y(z_1)] \leq 1 - f_y^1(\alpha_y),$$

where $f_y^1 = T(A_1(y), A_1'(y))$, and the inequality follows since $\phi_1^y$ is sub-optimal.

Now define $\phi_2^y : \mathcal{Z}_2 \to [0, 1]$ as the optimal level-$\alpha_y$ test for the problem $A_2(y)$ vs. $A_2'(y)$. Define the full test $\phi_2 : \mathcal{Y} \times \mathcal{Z}_2 \to [0, 1]$ by:

$$\phi_2(y, z_2) := \phi_2^y(z_2).$$

We now verify that $\phi_2$ has level $\alpha$:

$$\begin{aligned}
\mathbb{E}_{\zeta_2}[\phi_2] &= \mathbb{E}_{y \sim P}\left[\mathbb{E}_{z_2 \sim A_2(y)}[\phi_2^y(z_2)]\right] \\
&= \mathbb{E}_{y \sim P}[\alpha_y] \\
&= \mathbb{E}_{y \sim P}\left[\mathbb{E}_{z_1 \sim A_1(y)}[\phi_1^y(z_1)]\right] \\
&= \mathbb{E}_{\zeta_1}[\phi_1] = \alpha.
\end{aligned}$$

Next, we compute the power of $\phi_2$:

$$\begin{aligned}
\mathbb{E}_{\zeta_2'}[\phi_2] &= \mathbb{E}_{y \sim P'}\left[\mathbb{E}_{z_2 \sim A_2'(y)}[\phi_2^y(z_2)]\right] \\
&= \mathbb{E}_{y \sim P'}\left[1 - f_y^2(\alpha_y)\right] \quad \text{(since } \phi_2^y \text{ is optimal)} \\
&> \mathbb{E}_{y \sim P'}\left[1 - f_y^1(\alpha_y)\right] \quad \text{(by } f_y^1 \leq f_y^2) \\
&\geq \mathbb{E}_{y \sim P'}\left[\mathbb{E}_{z_1 \sim A_1'(y)}[\phi_1^y(z_1)]\right] \quad \text{(by sub-optimality of } \phi_1^y) \\
&= \mathbb{E}_{\zeta_1'}[\phi_1] = 1 - f^1(\alpha).
\end{aligned}$$

Thus, $\phi_2$ achieves the same level $\alpha$ but strictly greater power, completing the proof. $\qquad\square$

WELL-DEFINEDNESS OF $\otimes$

From definition, $f \otimes g := T(P \times P', Q \times Q')$ where $f = T(P,Q)$ and $g = T(P',Q')$. To show this is well-defined, suppose $f = T(P,Q) = T(P'',Q'')$; then it suffices to show:

$$T(P \times P', Q \times Q') = T(P'' \times P', Q'' \times Q').$$

**Lemma B.6.** *If $T(P,Q) \leq T(P'',Q'')$, then:*

$$T(P \times P', Q \times Q') \leq T(P'' \times P', Q'' \times Q').$$

*In particular, equality holds when $T(P,Q) = T(P'',Q'')$.*

*Proof of Lemma A.4.* If the algorithms output independently of $y$, then the joint distributions are products. Applying Lemma B.5 completes the proof. $\qquad\square$

Thus, $\otimes$ is well-defined, and satisfies:

$$g_1 \leq g_2 \Rightarrow f \otimes g_1 \leq f \otimes g_2.$$

TWO-STEP COMPOSITION

We now prove a compositional guarantee for two-step mechanisms. Before we proceed it is important to mention the all the influence is measured on $z_{test}$ and thus removed from the arguments of the algorithms.

**Lemma B.7.** *Let $S$ has $f$-influence for $A_1 : \mathcal{X} \to \mathcal{Y}$ and $g$-influence for $A_2(\cdot, y)$ for each $y \in \mathcal{Y}$ such that $A_2 : \mathcal{X} \times \mathcal{Y} \to \mathcal{Z}$. Then $S$ has is $(f \otimes g)$-influence for the composed mechanism $A(x) = A_2(x, A_1(x))$*

*Proof of Lemma A.5.* Let $Q, Q'$ be such that $g = T(Q, Q')$. Fix datasets $\mathcal{D} \setminus S$ and $\mathcal{D}$, and consider:

$$f_y^1 = T(A_2(\mathcal{D} \setminus S, y), A_2(\mathcal{D}, y)), \quad \forall y.$$

By the definition $f_y^1 \geq g$. Thus by Lemma B.5 the following holds:

$$\begin{aligned}
T(A(\mathcal{D} \setminus S), A(\mathcal{D})) &\geq T(A_1(\mathcal{D} \setminus S) \times Q, A_1(\mathcal{D}) \times Q') \\
&= T(A_1(\mathcal{D} \setminus S), A_1(\mathcal{D})) \otimes T(Q, Q') \\
&\geq f \otimes g.
\end{aligned}$$

Thus for the composed algorithm $A$, $S$ is $(f \otimes g)$-influential. $\qquad\square$

The above Lemma B.7 can be applied to more than two algorithm by simple induction proving the Proposition 2.6.

### B.3 COMPOSITIONALITY FOR GAUSSIAN INFLUENCE

**Corollary B.8.** *In the case of $G_\mu$-influence, for k-fold composition $G_{\mu_1} \otimes G_{\mu_2} \otimes \ldots \otimes G_{\mu_k} = G_\mu$ the following holds $\mu = \sqrt{\mu_1^2 + \ldots + \mu_k^2}$.*

Let $\mu = (\mu_1, \mu_2) \in \mathbb{R}^2$ and let $I_2$ denote the $2 \times 2$ identity matrix. Then we have:

$$\begin{aligned}
G_{\mu_1} \otimes G_{\mu_2} &= T\left(\mathcal{N}(0,1), \mathcal{N}(\mu_1, 1)\right) \otimes T\left(\mathcal{N}(0,1), \mathcal{N}(\mu_2, 1)\right) \\
&= T\left(\mathcal{N}(0,1) \times \mathcal{N}(0,1), \mathcal{N}(\mu_1, 1) \times \mathcal{N}(\mu_2, 1)\right) \\
&= T\left(\mathcal{N}(0, I_2), \mathcal{N}(\mu, I_2)\right).
\end{aligned}$$

We now use the invariance of trade-off functions under invertible transformations. The distribution $\mathcal{N}(0, I_2)$ is rotationally invariant, so we can apply a rotation to both distributions such that the mean vector becomes $(\sqrt{\mu_1^2 + \mu_2^2}, 0)$. Continuing the computation:

$$
\begin{aligned}
G_{\mu_1} \otimes G_{\mu_2} &= T\left(\mathcal{N}(0, I_2), \mathcal{N}(\mu, I_2)\right) \\
&= T\left(\mathcal{N}(0,1) \times \mathcal{N}(0,1), \mathcal{N}(\sqrt{\mu_1^2 + \mu_2^2}, 1) \times \mathcal{N}(0,1)\right) \\
&= T\left(\mathcal{N}(0,1), \mathcal{N}(\sqrt{\mu_1^2 + \mu_2^2}, 1)\right) \otimes T\left(\mathcal{N}(0,1), \mathcal{N}(0,1)\right) \\
&= G_{\sqrt{\mu_1^2 + \mu_2^2}} \otimes \mathrm{Id} \\
&= G_{\sqrt{\mu_1^2 + \mu_2^2}}.
\end{aligned}
$$

Thus $k-$fold composition will yield $\mu = \sqrt{\mu_1^2 + \ldots + \mu_k^2}$

### B.4 FUNCTIONALS OF $f$

As a preliminary step, we clarify the functionals $\nu_1, \nu_2, \nu_3, \bar{\nu}_3, \mu$ and $\gamma$ in Theorem B.12. We focus on symmetric trade-off functions $f$ with $f(0) = 1$, although many aspects of the discussion generalize beyond this subclass. Recall the definitions:

$$
\nu_1(f) = -\int_0^1 \log |f'(x)| \; dx; \quad \nu_2(f) = \int_0^1 \left(\log |f'(x)|\right)^2 \; dx; \quad \nu_3(f) = \int_0^1 |\log |f'(x)||^3 \; dx
$$

$$
\bar{\nu}_3(f) = \int_0^1 |\log |f'(x)| + \nu_1(f)|^3 \, dx, \quad \mu = \frac{2 \|\nu_1\|_1}{\sqrt{\|\nu_2\|_1 - \|\nu_1\|_2^2}} \quad \gamma = \frac{0.56 \|\bar{\nu}_3\|_1}{\left(\|\nu_2\|_1 - \|\nu_1\|_2^2\right)^{3/2}}
$$

We first confirm that these functionals are well-defined and take values in $[0, +\infty]$. For $\nu_2$ and $\bar{\nu}_3$, as well as the non-central version $\nu_3$, the integrands are non-negative, so the integrals are always well-defined (possibly infinite).

For $\nu_1$, potential singularities can occur at $x = 0$ and $x = 1$. If $x = 1$ is a singularity, then $\log |f'(x)| \to -\infty$ near 1, which is acceptable because the functional is permitted to take value $+\infty$. We must rule out the possibility that $\int_0^\varepsilon \log |f'(x)| \, dx = +\infty$ for some $\varepsilon > 0$. This cannot happen, since

$$
\log |f'(x)| \leq |f'(x)| - 1,
$$

and $|f'(x)| = -f'(x)$ is integrable on $[0, 1]$ because it is the derivative of $-f$, an absolutely continuous function. The non-negativity of $\nu_1(f)$ follows from Jensen's inequality. Dong et al. (2022) showed that

$$
\nu_1(T(P,Q)) = D_{\mathrm{KL}}(P \, \| \, Q),
$$

In fact, $\nu_2$ corresponds to another divergence known as the *exponential divergence*. We introduce a convenient notation for trade-off functions that will be useful in calculations below. For a trade-off function $f$, define

$$
D_f(x) := |f'(1-x)| = -f'(1-x),
$$

Using a simple change of variable, Dong et al. (2022) showed that we can rewrite these functionals as:

$$
\nu_1(f) = -\int_0^1 \log D_f(x) \, dx,
$$

$$
\nu_2(f) = \int_0^1 \left(\log D_f(x)\right)^2 \, dx,
$$

$$
\bar{\nu}_3(f) = \int_0^1 |\log D_f(x) + \nu_1(f)|^3 \, dx.
$$

The following shadows of the above functionals will appear in the proof:

$$\tilde{\nu}_1(f) = \int_0^1 Df(x) \log Df(x) \, dx$$

$$\tilde{\nu}_2(f) = \int_0^1 Df(x) \log^2 Df(x) \, dx,$$

$$\tilde{\nu}_3(f) = \int_0^1 Df(x)|\log Df(x) - \tilde{\nu}_1(f)|^3 \, dx.$$

These functionals are also well-defined on the space of trade-off functions $\mathcal{F}$ and take values in $[0, +\infty]$. The argument is similar to that used for $\nu_1$, $\nu_2$, and $\nu_3$. Dong et al. (2022) prove the following proposition:

**Proposition B.9.** *Suppose $f$ is a trade-off function and $f(0) = 1$. Then*

$$\tilde{\nu}_1(f) = \nu_1(f), \qquad \tilde{\nu}_2(f) = \nu_2(f), \qquad \tilde{\nu}_3(f) = \nu_3(f).$$

## B.5  PROOF OF NORMALITY IN NON-ASYMPTOTIC REGIME

**Lemma B.10.** *(normality boundedness) Let $f_1, \ldots, f_k$ be symmetric trade-off functions such that for some functionals $\nu_3, \mu, \gamma$ defined above assume, $\nu_3(f_i) < \infty, \forall i \in [k]$ and $\gamma < \frac{1}{2}$. Then $\forall \alpha \in [\gamma, 1 - \gamma]$, the following holds:*

$$G_\mu(\alpha + \gamma) - \gamma \le f_1 \otimes f_2 \otimes \ldots \otimes f_k(\alpha) \le G_\mu(\alpha - \gamma) + \gamma \tag{1}$$

Before we finally start the proof, let us recall the Berry–Esseen theorem for sums of random variables. Suppose we have $n$ independent random variables $X_1, \ldots, X_k$ with $\mathbb{E}(X_i) = \mu_i$, $\mathrm{Var}(X_i) = \sigma_i^2$, and $\mathbb{E}(|X_i - \mu_i|^3) = \rho_i$. Consider the normalized sum:

$$S_k := \frac{\sum_{i=1}^k (X_i - \mu_i)}{\sqrt{\sum_{i=1}^k \sigma_i^2}},$$

and let its cumulative distribution function (CDF) be $F_k$. Let $\Phi$ denote the standard normal CDF.

**Theorem B.11** (Berry–Esseen Theorem). *There exists a universal constant $C > 0$ such that*

$$\sup_{x \in \mathbb{R}} |F_k(x) - \Phi(x)| \le C \cdot \frac{\sum_{i=1}^k \rho_i}{\left(\sum_{i=1}^k \sigma_i^2\right)^{3/2}}.$$

*To the best of our knowledge, the best value of $C$ is $0.56$.*

*Proof.* For simplicity, let $f := f_1 \otimes f_2 \otimes \cdots \otimes f_k$. First, let us find distributions $P_0$ and $P_1$ such that $T(P_0, P_1) = f$. By symmetry, if $f_i(0) < 1$, then $f_i(x) = 0$ in some interval $[-\epsilon, \epsilon]$ for some $\epsilon > 0$, which yields $\nu_1(f_i) = +\infty$. So we may assume $f_i(0) = 1$ for all $i$.

Recall that $Df_i(x) = f_i(1 - x)$. Let $P$ be the uniform distribution on $[0, 1]$, and let $Q_i$ be the distribution on $[0, 1]$ with density $Df_i$. Since $f_i$ are symmetric and $f_i(0) = 1$, the supports of $P$ and all $Q_i$ are exactly $[0, 1]$, and we have $T(P, Q_i) = f_i$. Hence, by definition,

$$f = T(P^{\otimes k}, Q_1 \otimes \cdots \otimes Q_k)$$

Now let us study the hypothesis testing problem between $P^{\otimes k}$ and $Q_1 \otimes \cdots \otimes Q_k$. Let

$$L_i(x) := \log \frac{dQ_i}{dP}(x) = \log Df_i(x)$$

be the log-likelihood ratio for the $i$-th coordinate. Since both hypotheses are product distributions, the Neyman–Pearson lemma implies that the optimal rejection rule is a threshold function of the quantity $\sum_{i=1}^k L_i$. Further analysis of $\sum_{i=1}^k L_i(x_i)$ under both the null and alternative hypotheses; i.e., when $(x_1, \ldots, x_k)$ is drawn from $P^{\otimes k}$ or from $Q_1 \otimes \cdots \otimes Q_k$ is required.

To proceed we follow the exact steps by Dong et al. (2022). We first identify the quantities that exhibit central limit behavior, then express the test and $f(\alpha)$ in terms of these quantities.

For further simplification, let

$$T_k := \sum_{i=1}^{k} L_i.$$

As we suppress the $x_i$ notation, we should keep in mind that $T_k$ has different distributions under $P^{\otimes k}$ and $Q_1 \otimes \cdots \otimes Q_k$, though it is still a sum of independent random variables in both cases.

In order to find quantities with central limit behavior, it suffices to normalize $T_k$ under both distributions. The functionals Dong et al. (2022) introduced are specifically designed for this purpose.

$$\mathbb{E}_P[L_i] = \int_0^1 \log Df_i(x_i)\, dx_i = -\nu_1(f_i),$$

$$\mathbb{E}_{Q_i}[L_i] = \int_0^1 Df_i(x_i) \log Df_i(x_i)\, dx_i = \tilde{\nu}_1(f_i) = \nu_1(f_i),$$

Now lets define,

$$\mathbb{E}_{P^k}[T_k] = \sum_{i=1}^{k} -\nu_1(f_i) =: -||\nu_1||_1,$$

$$\mathbb{E}_{Q_1 \otimes \cdots \otimes Q_k}[T_k] = \sum_{i=1}^{k} \nu_1(f_i) = ||\nu_1||_1.$$

Similarly for the variances:

$$\mathrm{Var}_P[L_i] = \mathbb{E}_P[L_i^2] - \mathbb{E}_P[L_i]^2 = \mathrm{Var}_P[L_i] = \nu_2(f_i) - \nu_1^2(f_i),$$

$$\mathrm{Var}_{Q_i}[L_i] = \mathbb{E}_{Q_i}[L_i^2] - \mathbb{E}_{Q_i}[L_i]^2 = \nu_2(f_i) - \tilde{\nu}_1^2(f_i) = \nu_2(f_i) - \nu_1^2(f_i).$$

Therefore, the total variance under both hypotheses is:

$$\mathrm{Var}_{P^k}[T_k] = \mathrm{Var}_{Q_1 \otimes \cdots \otimes Q_k}[T_k] = \sum_{i=1}^{k} \left( \nu_2(f_i) - \nu_1^2(f_i) \right) =: ||\nu_2||_1 - ||\nu_1||_2^2.$$

In order to apply the Berry–Esseen Theorem (for random variables), we still need the centralized third moments:

$$\mathbb{E}_P\left[ |L_i - \mathbb{E}_P[L_i]|^3 \right] = \int_0^1 (\log Df_i(x) + \nu_1(f_i))^3\, dx =: \bar{\nu}_3(f_i),$$

$$\mathbb{E}_{Q_i}\left[ (L_i - \mathbb{E}_{Q_i}[L_i])^3 \right] = \int_0^1 Df_i(x) \left| \log Df_i(x) - \nu_1(f_i) \right) ||^3 dx = \tilde{\nu}_3(f_i) = \bar{\nu}_3(f_i).$$

Let $F_k$ be the CDF of the normalized statistic

$$\frac{T_k + ||\nu_1||_1}{\sqrt{||\nu_2||_1 - ||\nu_1||_2^2}} \quad \text{under } P^k,$$

and let $\tilde{F}^{(k)}$ be the CDF of

$$\frac{T_k - ||\nu_1||_1}{\sqrt{||\nu_2||_1 - ||\nu_1||_2^2}} \quad \text{under } Q_1 \otimes \cdots \otimes Q_k.$$

By Berry–Esseen Theorem, we have

$$\sup_{x \in \mathbb{R}} |F_k(x) - \Phi(x)| \leq C \cdot \frac{||\nu_3||_1}{(||\nu_2||_1 - ||\nu_1||_2^2)^{3/2}}, \tag{27}$$

and similarly for $F^{(k)}$.

So we have identified the quantities that exhibit central limit behavior.

Now let us relate them with $f$. Consider the testing problem $(P^k, Q_1 \otimes \cdots \otimes Q_k)$. For a fixed $\alpha \in [0, 1]$, let $\phi$ be the (potentially randomized) optimal rejection rule at level $\alpha$. By the Neyman–Pearson lemma, $\phi$ must threshold $T_k$.

An equivalent form that highlights the central limit behavior is the following:

$$\phi = \begin{cases} 1 & \text{if } \frac{T_k + \|\nu_1\|_1}{\sqrt{\|\nu_2\|_1 - \|\nu_1\|_2^2}} > t, \\ p & \text{if } \frac{T_k + \|\nu_1\|_1}{\sqrt{\|\nu_2\|_1 - \|\nu_1\|_2^2}} = t, \\ 0 & \text{otherwise,} \end{cases}$$

where $t$ and $p \in [0, 1]$ are chosen to achieve size $\alpha$.

Let $t \in \mathbb{R} \cup \{\pm\infty\}$ and $p \in [0, 1]$ be parameters uniquely determined by the condition $\mathbb{E}_{P^k}[\varphi] = \alpha$. With this, the expectation under $P^k$ can be written in terms of the empirical CDF $F_k$ as:

$$\mathbb{E}_{P^k}[\varphi] = P^k \left[ T_k + \frac{\|\nu_1\|_1}{\sqrt{\|\nu_2\|_1 - \|\nu_1\|_2^2}} > t \right] + p \cdot P^k \left[ T_k + \frac{\|\nu_1\|_1}{\sqrt{\|\nu_2\|_1 - \|\nu_1\|_2^2}} = t \right]$$
$$= 1 - F_k(t) + p \cdot [F_k(t) - F_k(t^-)],$$

where $F_k(t^-)$ is the left limit of $F_k$ at $t$. A simple rearrangement gives:

$$1 - \alpha = 1 - \mathbb{E}_{P^k}[\varphi] = (1 - p)F_k(t) + pF_k(t^-),$$

and hence the inequality

$$F_k(t^-) \leq 1 - \alpha \leq F_k(t).$$

Now consider $\mathbb{E}_{Q_1 \times \cdots \times Q_k}[\varphi]$. It is helpful to define an auxiliary variable $\tau := t - \mu$, where $\mu$ was defined in the theorem statement as:

$$\mu := \frac{2\|\nu_1\|_1}{\sqrt{\|\nu_2\|_1 - \|\nu_1\|_2^2}}.$$

This gives the equivalence:

$$T_k + \frac{\|\nu_1\|_1}{\sqrt{\|\nu_2\|_1 - \|\nu_1\|_2^2}} > t \quad \Longleftrightarrow \quad T_k - \frac{\|\nu_1\|_1}{\sqrt{\|\nu_2\|_1 - \|\nu_1\|_2^2}} > \tau. \tag{28}$$

Using this, we can express:

$$1 - f(\alpha) = \mathbb{E}_{Q_1 \times \cdots \times Q_k}[\varphi]$$

$$= Q_1 \times \cdots \times Q_k \left[ T_k + \frac{\|\nu_1\|_1}{\sqrt{\|\nu_2\|_1 - \|\nu_1\|_2^2}} > t \right]$$

$$+ p \cdot Q_1 \times \cdots \times Q_k \left[ T_k + \frac{\|\nu_1\|_1}{\sqrt{\|\nu_2\|_1 - \|\nu_1\|_2^2}} = t \right]$$

$$= Q_1 \times \cdots \times Q_k \left[ T_k - \frac{\|\nu_1\|_1}{\sqrt{\|\nu_2\|_1 - \|\nu_1\|_2^2}} > \tau \right]$$

$$+ p \cdot Q_1 \times \cdots \times Q_k \left[ T_k - \frac{\|\nu_1\|_1}{\sqrt{\|\nu_2\|_1 - \|\nu_1\|_2^2}} = \tau \right]$$

$$= 1 - \widetilde{F}^{(k)}(\tau) + p \cdot [\widetilde{F}^{(k)}(\tau) - \widetilde{F}^{(k)}(\tau^-)],$$

where $\widetilde{F}^{(k)}$ is the CDF under $Q_1 \times \cdots \times Q_k$. Rearranging gives:

$$f(\alpha) = (1 - p) \cdot \widetilde{F}^{(k)}(\tau) + p \cdot \widetilde{F}^{(k)}(\tau^-),$$

and thus the inequality:

$$\widetilde{F}^{(k)}(\tau^-) \le f(\alpha) \le \widetilde{F}^{(k)}(\tau).$$

So far we have:

$$F_k(t^-) \le 1 - \alpha \le F_k(t), \tag{29}$$

$$\widetilde{F}^{(k)}(\tau^-) \le f(\alpha) \le \widetilde{F}^{(k)}(\tau). \tag{30}$$

From inequality (27), we know that both $F_k$ and $\widetilde{F}^{(k)}$ are $\gamma$-close to the standard normal CDF $\Phi$, so:

$$\Phi(t) - \gamma \le F_k(t^-) \le 1 - \alpha \le F_k(t) \le \Phi(t) + \gamma,$$

which implies:

$$\Phi^{-1}(1 - \alpha - \gamma) \le t \le \Phi^{-1}(1 - \alpha + \gamma). \tag{31}$$

Using (30) and (31), we can upper-bound $f(\alpha)$:

$$\begin{aligned}
f(\alpha) &\le \widetilde{F}^{(k)}(\tau) \\
&\le \Phi(\tau) + \gamma \\
&= \Phi(t - \mu) + \gamma \\
&\le \Phi(\Phi^{-1}(1 - \alpha + \gamma) - \mu) + \gamma \\
&= G_\mu(\alpha - \gamma) + \gamma.
\end{aligned}$$

Similarly, we obtain the lower bound:

$$f(\alpha) \ge G_\mu(\alpha + \gamma) - \gamma.$$

This completes the proof. $\qquad \square$

## B.6 Proof of Theorem 2.8

**Theorem B.12.** *(asymptotic normality) Let $\{f_{ki} : i \in [k]\}_{k=1}^\infty$ be a triangular array of symmetric trade-off functions and for some functionals $\nu_1, \nu_2, \nu_3$, $M \ge 0$ and $s > 0$, assume $\sum_{i=1}^k \nu_1(f_{ki}) \to M$, $\max_{1 \le i \le k} \nu_1(f_{ki}) \to 0$, $\sum_{i=1}^k \nu_2(f_{ki}) \to s^2$, $\sum_{i=1}^k \nu_3(f_{ki}) \to 0$. Then the following holds:*

$$\lim_{k \to \infty} f_{k1} \otimes \ldots \otimes f_{kk}(\alpha) = G_{2M/s}(\alpha) \tag{2}$$

*Proof.* We first establish pointwise convergence $f_{k1} \otimes \cdots \otimes f_{kk} \to G_{2M/s}$, and then deduce uniform convergence using a general theorem.

By Lemma B.10, applied to the $k$-th row of the triangular array, we get

$$G_{\mu_k}(\alpha + \gamma_k) - \gamma_k \le f_{k1} \otimes \cdots \otimes f_{kk}(\alpha) \le G_{\mu_k}(\alpha - \gamma_k) + \gamma_k,$$

where

$$\mu_k = \frac{2\|\nu_1^{(k)}\|_1}{\sqrt{\|\nu_2^{(k)}\|_1 - \|\nu_1^{(k)}\|_2^2}}, \quad \gamma_k = 0.56 \cdot \frac{\|\bar{\nu}_3^{(k)}\|_1}{(\|\nu_2^{(k)}\|_1 - \|\nu_1^{(k)}\|_2^2)^{3/2}}.$$

We will show that $\mu_k \to 2M/s$ and $\gamma_k \to 0$. The assumptions imply:

$$\|\nu_1^{(k)}\|_1 \to M, \quad \|\nu_1^{(k)}\|_\infty \to 0, \quad \|\nu_2^{(k)}\|_1 \to s^2, \quad \|\nu_3^{(k)}\|_1 \to 0.$$

First, observe

$$\|\nu_1^{(k)}\|_2^2 = \langle \nu_1^{(k)}, \nu_1^{(k)} \rangle \le \|\nu_1^{(k)}\|_\infty \cdot \|\nu_1^{(k)}\|_1 \to 0.$$

To bound $\|\bar{\nu}_3^{(k)}\|_1$, we use the following lemma from Dong et al. (2022):

**Lemma B.13.** *For any trade-off function $f$, we have*

$$\bar{\nu}_3(f) \leq \nu_3(f) + 3\nu_1(f)\nu_2(f) + 3\nu_1(f)^2\sqrt{\nu_2(f)} + \nu_1(f)^3.$$

Applying the lemma to each $f_{ki}$, summing and using Cauchy-Schwarz inequality ($|\sum_i a_i b_i| \leq |\sum_i a_i| \cdot \max |b_i|$), we get:

$$\|\bar{\nu}_3^{(k)}\|_1 \leq \|\nu_3^{(k)}\|_1 + 3\|\nu_1^{(k)}\|_\infty\|\nu_2^{(k)}\|_1 + 3\|\nu_1^{(k)}\|_\infty\sqrt{\|\nu_2^{(k)}\|_1 \cdot \|\nu_1^{(k)}\|_2^2} + \|\nu_1^{(k)}\|_\infty^2\|\nu_1^{(k)}\|_1 \to 0.$$

Therefore, $\mu_k \to 2M/s$ and $\gamma_k \to 0$ as by assumptions $\|\nu_1^{(k)}\|_1 \to M$, $\|\nu_1^{(k)}\|_\infty \to 0$, $\|\nu_2^{(k)}\|_1 \to s^2$, $\|\nu_3^{(k)}\|_1 \to 0$, and $\|\nu_1^{(k)}\|_2^2 \to 0$. Since $G_\mu(\alpha)$ is continuous in both $\alpha$ and $\mu$, we conclude

$$G_{\mu_k}(\alpha \pm \gamma_k) \pm \gamma_k \to G_{2M/s}(\alpha),$$

which proves pointwise convergence.

For boundary points, note that $\alpha = 0$ implies $G_{\mu_k}(0 + \gamma_k) - \gamma_k \to 1 = G_{2K/s}(0)$, and similarly at $\alpha = 1$. Now we use the following lemma from Dong et al. (2022), which is stated here for completeness.

**Lemma B.14.** *Let $\{f_n\} : [a, b] \to \mathbb{R}$ be a sequence of non-increasing functions. If $f_n$ converges pointwise to a function $f : [a, b] \to \mathbb{R}$ and $f$ is continuous on $[a, b]$, then the convergence is uniform.*

Finally, uniform convergence follows from the above lemma, producing the desired result. □

## C  IDENTIFYING MISLABELED SAMPLES IN CIFAR-10

To further prove the utility of our method for higher-dimensional settings, we follow the same setup as in section 4.2 on CIFAR-10 (Krizhevsky et al., 2009) dataset using a ResNet-18 model. From Figure 12, we observe that our method achieves performance comparable to TraceIn. On average, it outperforms TRAK and Influence Functions by 10.21% and 14.04%, respectively, in this setting. For this experiment, we report the mean recall values over three random training runs with f-INE achieving the lowest variance of 0.01, whereas TraceIn has a variance of 0.02, TRAK has a variance of 0.03, and Influence Function achieves a variance of 0.02. Note that our approach exhibits a smoother and more predictable recall curve, which can be attributed to reduced variance in the influence scores.

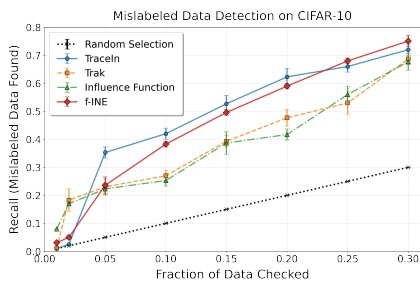

Figure 12: Utility of f-INE for finding mislabeled samples on the CIFAR-10 dataset.

## D  QUALITATIVE CASE STUDY ON MODEL EXPLAINABILITY

The primary objective of influence-estimation techniques is to identify the most influential training samples for a given test instance. Figure 13 presents a qualitative evaluation of our method on mini-ImageNet (Huh et al., 2016) dataset using a ResNet-50 model. As shown, for a selected test sample, our approach consistently assigns the highest influence scores to semantically coherent examples within the same class.

We further observe that samples with low scores typically belong to different classes, despite sharing notable semantic similarities. This behavior is intuitively reasonable as training samples that are semantically similar yet originate from different classes are generally considered harmful for the prediction of the given test input.

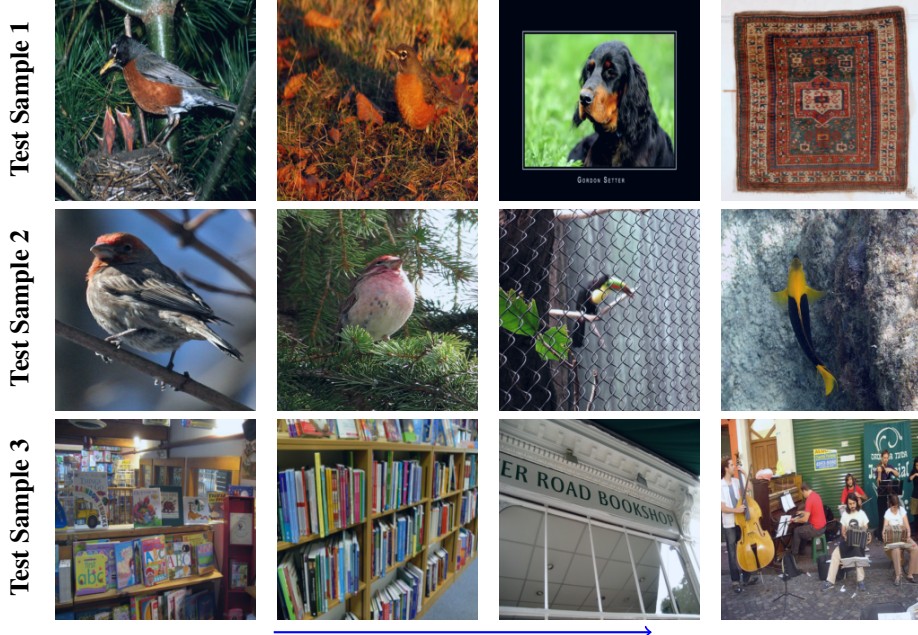

Figure 13: For each test sample shown in the left column, the second through fourth columns display training samples sorted in terms of descending influence scores. We observe that our method consistently assigns the highest influence to semantically coherent, same-class examples. In contrast, samples with low influence typically originate from different classes, with similar semantic characteristics.

## E    ADDITIONAL IMPLEMENTATION DETAILS

• **Codebase:** For reproducibility, we have included a preliminary version of our code here: `https://github.com/Subhodip123/f-INE`. We will keep updating this codebase with more improvements.

• **Training of LLMs** We use LoRA (Hu et al., 2022) to efficiently finetune Llama-3.1-8B on the poisoned LIMA dataset for 15 epochs using the same setup and hyperparameters as Zhou et al. (2023). We save model states across 100 equally spaced checkpoints throughout the training run to collect gradients for influence estimation. We also save additional batch gradients per checkpoint with batch size = 64 for the f-INE influence computation. Following Xia et al. (2024), we apply random projections to store the LoRA gradients with $d = 8192$ for memory efficiency. We replicate training across 3 random seeds.

• **Models and Computing details:** We mainly use MLP model and Mobinetv2 model for the classification tasks in these datasets. Our MLP model has only one hidden dimension of size 500. We train this MLP model from scratch on a single NVIDIA A-6000 (48 GB) GPU, achieving test accuracy of 97% MNIST dataset. MobileNetV2 is a lightweight and efficient convolutional neural network architecture consisting of residual blocks, linear bottlenecks, and depth-wise separable convolution layers. For training this model, we use the ImageNet pre-trained model weights and change the last layer size based on the classification task. We finetuned the whole model on the downstream datasets on the same GPU.

• **Hyper-parameter Details:** We have trained all the models for $T = 100$ epochs with batch size of 100. We have used Adam optimizer with learning rate $\eta = 0.005$, $\beta_1 = 0.9$ and $\beta_2 = 0.99$. We have used cross-entropy loss for all the classification tasks.

