# OpenReview forum: "f-INE: A Hypothesis Testing Framework for Estimating Influence under Training Randomness"
_ICLR.cc/2026/Conference — ICLR 2026 Poster_

### Official Review · Reviewer_EzUe · 2025-10-29

**Soundness:** 3
**Presentation:** 3
**Contribution:** 3
**Rating:** 6
**Confidence:** 3

**Summary:**

In this paper the authors propose a novel data influence estimation algorithm which is an improvement on top of the existing TracIn (Training Data Influence by Tracing Gradient Descent) approach. The proposed extension is less vulnerable to random training runs, and is motivated by privacy auditing and hypothesis testing theories. The paper estimates the statistical significance of data influence by quantifying the distinguishability of two distributions: a distribution of influence scores when the sample participates in training over multiple random training trials and a distribution when the sample did not participate in the training process. The paper proposes to look at the tradeoff curves between Type I and Type II errors as a means to quantify the distinguishability of those two distributions.
It draws parallels between their definition and Gaussian Differential Privacy, claiming that the tradeoff can be fully characterized by a single variable, mean.
The paper shows that the proposed approach is more stable and is effective for MNIST's miss-labeling task and train-test influence for LLM’s data poisoning tasks.

**Strengths:**

+ The paper is well and clearly written without major ambiguities.
+ The paper studies an important problem, randomness in the influence estimation. It proposes an interesting approach motivated by the Differential Privacy theory and explains the theoretical intuition behind it.

**Weaknesses:**

+ See weaknesses
+ In Figure 7 the error bar for F-INF looks pretty large. If  F-INF is more stable than LESS why does it have wider std ?

**Questions:**

+ The experimental evaluation part is a bit lightweight
  + MNIST is a very toy model and LLM experiments are good but cover a narrow set of applications -  bias in a specific category.
It is unclear how well the approach performs for a wider range of datasets or models. E.g. LESS performs the evaluation on a wider range of downstream tasks and datasets (bbh, tinyqa and mmlu). In addition to that, TRAK evaluates their approach on fact checking dataset. It would be good to report results on a wider range of tasks and datasets similar to those baseline approaches.
+ The algorithm description is a bit unclear
	+ E.g. Algorithm1: lines 9 and 10 the right side of the assignment seem to be identical,
               likely there is a typo there. Similarly line 4 and line 5 perform the same sampling (D \ S ) ?
+ It is unclear how S is selected in Algorithm listing 1 and whether Algorithm 1 is called in a loop for random subsets of S. Is it guaranteed that each example in S will be encountered only once in S ?
+ It seems that instead of retraining the model multiple times the authors leverage training epochs but this can be a very noisy approximation.

**Details Of Ethics Concerns:**

N /  A

---

> ### Author Response · Authors · 2025-11-20
> **Rebuttal by Authors**
>
> We thank the reviewer for the positive comments about the paper. Specific doubts are clarified below:
>
> 1. **More Experimental evaluations**
> - To show the efficacy of our method in relatively high-dimensional settings, we have experimented with finding mislabeled samples on CIFAR-10 data sets using Resnet-18 model. The table below shows that our method is comparable with TraceIN while outperforming TRAK by 10.21% and Influence Function by 14.04% on average, with lower variations. These results are added in Appendix B of the new version of the PDF.
>
>     | % Data Checked | TraceIn | Trak | Influence Function | Our Method |
>     |--------------------------|---------|------|---------------------|------------|
>     | 1%                       | 1%      | 12%   | 8%                   | 3.1%       |
>     | 2%                       | 2.5%    | 18.3% | 17%                   | 5%         |
>     | 5%                       | 35.3%   | 23%   | 22.3%                  | 23.6%      |
>     | 10%                      | 42%     | 27%   | 25.3%                  | 38.3%      |
>     | 15%                      | 52.7%   | 39.3% | 38.7%                  | 49.6%      |
>     | 20%                      | 62.3%   | 47.7% | 41.7%                  | 59%        |
>     | 25%                      | 66%     | 53%   | 56%                     | 68%        |
>     | 30%                      | 72%     | 68.7% | 67.7%                  | 75.1%      |
>
> - Additionally, to show model's explainability, we have added an additional case study where we use mini-ImageNet dataset (subset of ImageNet Dataset with 100 classes) with Resnet-50 model. The results can be found in Appendix section C of the new version of the PDF.
> - For further downstream evaluation, we have also tried to replicate the Fact tracing experiment as mentioned in TRAK. However, due to the high computation complexity of TRAK method (as mentioned in Table-1) we are unable to scale it to 8B Llamma 3.1 model. Further, the Llamma 8B already has very high accuracy (.94+) on the FTRACE-TREX dataset even without any training, and so the additional finetuning does not have a significant influence to trace. Hence, we are instead trying to replicate this experiment using our f-INE method with lower-dimensional mT5 model which was used in the TRAK paper. However, due to architectural differences, this will take a significant time to re-engineer our codebase, and hence, we are unable to provide such results due to the short rebuttal period. We will be sure to include them in the final version of the paper and will also update the results here if we get them in time.
>
>
> 2. **Clarification on Algorithm 1**
> - Yes, there is a typo in Line 9. In line 9 the right-hand should be SGD update with $\theta^t$ not $\hat{\theta}^t$. We thank the reviewer for pointing this out. However, lines 4 and 5 are correct. There are two data batches are samples one $B_t$ and another is $B'_t$. As we want the two distributions in $O$ and $O'$ to have low inter-dependence, that is why two random batches are sampled. This is a common technique that is usually applied in privacy literature[2]. We have updated our PDF.
>
>
> 3. **How to select S?**
> - Note that Algorithm 1 is written in a generic form. Here, the user is interested in estimating the influence of a particular data subset $S$. Thus, the algorithm takes $S$ as the user input. If one is interested in estimating influence of one particular sample, then $S=\{z_i\}$ for any particular $z_i \in \mathcal{D}$.
>
>
> 4. **Training epochs can be a very noisy approximation.**
> - We acknowledge that estimating influence using training epochs is noisy and in fact, one of the limitations. However, we would like to point out that, in spite of this pragmatic step also being employed in previous works [1,3], we have a solid theoretical composition result (asymptotic normality Theorem 2.8) which implies that in spite of noisy approximation from a training epoch, the final estimate converges if $T$ is large. Further, this step is practical/essential in order to reduce the computational cost of retraining multiple times, which is problematic for high-dimensional models such as LLMs.
>
> [1] Garima et al. Estimating training data influence by tracing gradient descent. In Proceedings of NeurIPS, 2020.
>
> [2] Milad Nasr et al. Tight auditing of differentially private machine learning. Proceedings of the 32nd USENIX Conference on Security Symposium, pp. 1631 – 1648, 2023
>
> [3] Wang et al. DATA SHAPLEY IN ONE TRAINING RUN, In Proc of ICLR, 2025

---

> > ### Comment · Reviewer_EzUe · 2025-11-27
> >
> > Thank you very much for replying to my feedback!
> > 1. Thank you for comparing your method with other baselines that I mentioned. It is interesting that Influence functions(second order approximation) does so poorly compared to tracIn. Do you know why it does so poorly on the mislabeling task ?
> > Also looks like TracIn outperforms f-INE except for the 25% and 30% data fractions. Any ideas why TracIn does better in those cases?
> > 2. Also I had this other question
> > In Figure 7 (Abortion) the error bar for F-INE looks pretty large. If F-INE is more stable than LESS why does it have wider std ?

---

> ### Author Response · Authors · 2025-11-29
> **Clarification on Typo in the table**
>
> We thank the reviewer for participating in the discussion.
> - We should point out that while writing the table on OpenReview there is a typo that has happened in the above table for mislabeling the experiment in CIFAR-10. For Influence Function, in the first and second row of the above table, while converting from fraction to % we have mistyped it. It is not 0.8% but rather 8% and not 1.7% but 17%. We have fixed this typo in the above table and PDF. We apologize for the typo.
> - Note that the performances of TRAK and the Influence Function are similar. This trend is also seen for both MNIST and CIFAR-10 mislabeling experiments.
>
> However, your questions remain still valid. We will address this in a detailed manner in our subsequent comment.

---

> ### Author Response · Authors · 2025-12-03
> **Reply to Further Comments by Authors**
>
> 1. **Reason for poor performance of Influence Function**
> - The primary reason for the poor performance is that, in high-dimensional neural networks, influence functions fail to accurately approximate the true leave-one-out (LOO) effect. As a consequence, they become ineffective at identifying mislabeled or corrupted training samples. This limitation has also been highlighted in prior works [1,2].
> - Furthermore, note that the influence estimated by influence functions is computed as a Hessian-preconditioned gradient inner product between the test point and a training point. In practice, evaluating the inverse Hessian of a modern neural network is computationally prohibitive, and thus it is typically approximated using conjugate-gradient-based solvers or stochastic estimation techniques. As the dimensionality of the model increases, these approximations become increasingly inaccurate, leading to larger estimation errors and ultimately degrading the performance of the influence function.
>
> 2. **Why TraceIN is better in other cases except 25% and 30% data fractions?**
> - Note that f-INE also outperforms TraceIN for 1% and 2% data cases. However, this is a very interesting question.
> -  We are absolutely not sure but hypothesize that accounting for the whole distribution f-INE is able to capture long tail data influences (see the case study on LLM in section 4.5). In this mislabeling experiment settings for CIFAR-10 dataset, we estimate the self-influence i.e. influence of a sample on the prediction on itself. It might happen that as certain subclasses are present in long tails of data, these samples exert higher self-influence because they are essentially memorized by the model[3]. Thus, in cases for 5%-20% data cases, these samples (apart from the mislabeled ones) appear which are captured by f-INE.
>
> 3. **Variance on Figure 7**
> - Note that Figure 8 shows the variance in the actual influence scores across seeds, f-INE scores have lower variance in all cases.
> - The variance in Figure 7 reflects variance in rankings -- to find the poisonded samples we sort the influence scores in descending order. However reducing this ranking can be an interesting research direction for the future.
>
> [1] Basu, S., Pope, P., and Feizi, S. Influence functions in deep learning are fragile. In Proc. of ICLR (2021).
>
> [2] Bae, J. et al. . If influence functions are the answer, then what is the question? In Proc. of NeurIPS (2022).
>
> [3] Vitaly Feldman et al. What Neural Networks Memorize and Why:
> Discovering the Long Tail via Influence Estimation. In Proc. of NeurIPS (2020)

---

### Official Review · Reviewer_ap96 · 2025-10-31

**Soundness:** 2
**Presentation:** 1
**Contribution:** 2
**Rating:** 2
**Confidence:** 3

**Summary:**

hey This paper identifies the sensitivity of influence estimation methods to training randomness (e.g., data ordering) as a critical flaw. It proposes f-INE, a new framework grounded in hypothesis testing, to provide an influence estimate that is robust to this randomness. The authors apply this method to mislabeled data detection on MNIST and to attributing behavior in an LLM.

While the technical idea is interesting, the paper is built on a flawed premise that mischaracterizes the research landscape. It ignores two distinct, highly relevant lines of established research that either (1) value the training-run-specific influence the authors dismiss as a "flaw" or (2) have already provided solutions for the exact problem the authors claim to be solving. The empirical results are invalid and do not support the claims.

**Strengths:**

The technical framework itself, which uses hypothesis testing to quantify the distinguishability between training runs, has some merit. This approach could potentially be valuable for other problems, even if it is misapplied here.

The layout of the manuscript looks comfortable and the manuscript is easy to read.

**Weaknesses:**

1. The paper's central premise—that robustness to training data ordering is universally desirable and that sensitivity to it is a "flaw"—is incorrect. There is an established line of research (e.g., [Data Shapley in One Training Run]) dedicated to quantifying a sample's contribution within a specific training run. This research finds that a sample's influence varies significantly at different training stages, guilding decisions on what data to train on during different stages.

2. The paper aims to find a data-centric utility measure that is robust to training randomness. Yet it fails to cite or discuss the large, established field of "data valuation" which also applies to this problem. Methods such as Data Shapley (https://arxiv.org/abs/1902.10275 ), LAVA (https://arxiv.org/abs/2305.00054 ) provide metrics for data utility from a pure data perspective, free from the randomness of any single training run. Data Banzhaf (https://arxiv.org/abs/2205.15466) is specifically designed to maximize robustness to training variations. The omission of this entire body of work is a critical flaw.

3. The experiments are insufficient to support the paper's claims. In the mislabeled MNIST experiment, the baselines appear to be incorrectly chosen or implemented. The results show that f-INE performs similarly to other methods, failing to demonstrate its purported superiority. The second experiment (LLM attribution) is conducted on a sample size (training on 50 examples, testing on 10) so tiny that it is impossible to derive any scientifically valid conclusions. Furthermore, this experiment only compares f-INE to LESS. Citing LESS as the sole baseline is questionable, as it was proposed for large-scale data selection, not the small-scale setting tested here.

**Questions:**

Can the authors explain a bit on how should we interpret the results from the MNIST experiment where all the methods perform similarly?

---

> ### Author Response · Authors · 2025-11-20
> **Part-1 Rebuttal by Authors**
>
> We thank the reviewer for the feedback. Specific doubt's are clarified below:
>
> 1. **Purported flaw in the central question**
> - We are surprised by the reviewer's claim and are not sure what they mean - Figure 1 already lays out why sensitivity to training randomness is bad.
> - Perhaps the reviewer is thinking about using influence estimation as an interpretability tool rather than a decision making tool (we focus on the latter).
> - In interpretability, one might be interested in asking influence score specific to one training data order/randomness. This is only a post-hoc analysis of what happened.
> - However, making a decision based on those scores might not translate to the next run. For example, in our experimental setting of identifying poisoned/corrupted samples to improve the model, one is required to guarantee that removing poisoned samples will change our training model in predictable ways. This concern is further reiterated in previous works [1,2]. **In TRAK[1] authors mention that "Non-determinism poses a problem for data attribution because by definition, we cannot explain such seed-based differences in terms of the training data." This is why robustness to randomness is an important characteristic in such settings.**
> - If randomness stems mostly from training data ordering, can we simply freeze the order? This is highly impractical - the data curator and data consumer are likely different entities. If I am cleaning up my data to make it public, I can hardly force everyone who uses the dataset to use a fixed ordering.
> - The reviewer also hints at an interesting research direction - defining different influence scores for different stages of the training run (early, mid, late). This is however, orthogonal to the point about randomness because even within each stage, what data shows up is a random variable in a given single training run and may be different in the next run.
>
> 2. **Discussion on "data valuation" and important citations missing**
> - Our related work section in the Appendix discusses data valuation methods (including Data Shapley[3]) and the distinction between Data Valuation and Influence Estimation. We are assuming that the reviewer might have missed this part and request the reviewer to refer to the lines 645-653 in the older version of the PDF and 712-730 in the updated version of the PDF.
> - However, we agree that it could be improved and expanded. We have slightly expanded it and now include LAVA[4] and Data Banzhaf[5] along with [7] in our updated version. We would be happy to add any other work the reviewer recommends to discuss.
> - We would like to highlight that while Data-banzaf is motivated by randomness, not part of the formalism. Our question about how should we define influence in presence of training randomness has not been addressed by these.
> - Finally, and most importantly, our focus is on LOO influence estimation and not general Data Valuation. We are **not trying to solve data valuation**. Our question is the much simpler - how do we measure the marginal contribution of $S$ i.e. can we predict what will happen when I delete $S$ and retrain the model given that the training is random? As we mentioned in the conclusion, our approach to dealing with training randomness can likely be extended to Data Shapley etc., which look at all marginal contributions.
>
>
>
> [1] Park et al. Trak: Attributing model behavior at scale. In Proc. of ICML, 2023
>
> [2] Karthikeyan et al. Revisiting methods for finding influential examples. In Proc. of AAAI, 2022
>
> [3] Amirata Ghorbani et al: Equitable valuation of data for machine learning. In Proc. of ICML, 2019.
>
> [4] Just et al. LAVA: DATA VALUATION WITHOUT PRE-SPECIFIED LEARNING ALGORITHMS, In Proc. of ICLR,2023
>
> [5] Wang et al. Data Banzhaf: A Robust Data Valuation Framework for Machine Learning, arXiv Preprint, 2023
>
> [6] Pang Wei Koh and Percy Liang. Understanding black-box predictions via influence functions. In Proc. of ICML, 2017
>
> [7] Wang et al. DATA SHAPLEY IN ONE TRAINING RUN, In Proc of ICLR, 2025

---

> ### Author Response · Authors · 2025-11-20
> **Part-2 Rebuttal by Authors**
>
> 3. **Experimental Flaws and Interpretation**
> - Note that we chose standard data attribution baselines with official implementations. In MNIST mislabeled experiments, a higher curve indicates a superior method. In this setting, on average, our method performs slightly better than TraceIN (only 0.05% better). However, our method performs 13.85% better than TRAK and 3.83% better than Influence Function. Please refer to lines 377-381.
> - Further to show our method's utility in a relatively higher-dimensional setting, we have added additional experiments of finding mislabeled samples in CIFAR-10 dataset using ResNet-18 model. Our results show that our method is comparable with TraceIN while outperforming TRAK by 10.21% and Influence Function by 14.04% on average, with lower variations. These results are added in Appendix B of the new version of the PDF.
> - The reviewer seems to have also misunderstood the setup of the LLM experiment. We instruction-tune the model on the training set LIMA, which has 1000 instances, and poison it with 50 samples per entity. It is trained on the whole dataset. The test set is to ascertain the LLM sentiment i.e., understand how the poisoned samples affected the **LLM behavior**. This is a standard instruction tuning setup as seen in [8].
> - For the LLM setting, we compare FINE to LESS since, as shown in Table 1, LESS is the only method that scales to models of the size of 8 billion parameters. For example, TRAK scales linearly with the number of ensemble models (retrainings) and is quadratic with respect to the model dimensions.
>
> [8] Zhou et al. LIMA: Less Is More for Alignment, In Proc of NeurIPS, 2023

---

### Official Review · Reviewer_V6TH · 2025-11-03

**Soundness:** 3
**Presentation:** 2
**Contribution:** 3
**Rating:** 6
**Confidence:** 4

**Summary:**

This paper introduces F-INE, a framework to estimate data influence that accounts for training randomness. This is important and interesting problem, and how the authors addressed this is also interesting. The core idea is to reframe influence as a hypothesis testing problem, which the authors connect to membership inference. They show this f-influence converges to a single scalar ($G_{\mu}$) for composed algorithms like SGD. The paper proposes an efficient, single-pass algorithm (f-INE) to estimate this $G_{\mu}$. Empirically, F-INE is shown to be significantly more consistent and robust than baselines, while demonstrating superior utility in detecting mislabeled data (MNIST) and poisoned instructions in Llama-3.1-8B.

**Strengths:**

1. The paper's focus on tackling training randomness is an important and necessary direction for the field. This is a well-known, critical flaw in existing influence estimation methods, and the paper addresses it head-on.

2. The theoretical connection established between influence estimation and hypothesis testing (specifically, membership inference and f-DP) is interesting. This provides a principled and statistically grounded foundation for the proposed method.

3. The empirical results are impressive. The high consistency score on the MNIST/MLP setup and the demonstration of superior stability and utility on the Llama-3.1-8B scale validate the method's practical advantages over existing SOTA baselines.

**Weaknesses:**

1. My main concern is the limited scope of the mislabeled data detection experiments. While the MNIST/MLP results are promising for showing consistency, this is a relatively simple setting. The claim of the method's general utility for data cleanup would be strengthened by evaluating it on more complex vision datasets like CIFAR-10/100 or Tiny-ImageNet, and on different architectures (e.g., ResNet family). I wonder if the strong performance and stability guarantees hold in these more challenging, higher-dimensional settings.

2. Another concern is the separation of the utility and consistency metrics. While both are important, they don't fully answer the key practical question: how consistently are the correct (ground-truth) items identified? These two metrics need to be simultaneously measured, perhaps as a conditional metric. The paper would be stronger if it included a unified metric. This would directly measure if the method is reliably useful.

**Questions:**

Please see the Weaknesses.

---

> ### Author Response · Authors · 2025-11-20
> **Rebuttal by Authors**
>
> We thank the reviewer for the positive comments about the paper. Specific doubts are clarified below:
>
> 1. **Data Clean up experiments on high dimensional settings**
>
> - As per the reviewer's suggestion to showcase our method's efficacy in relatively high-dimensional settings, we mislabel 20% of the CIFAR-10 data and used Resnet-18 model. The table below shows that our method is comparable with TraceIN while outperforming TRAK by 10.21% and Influence Function by 14.04% on average with lower variation. All of the results have similar trends as we expected, with f-INE matching or outperforming other methods.
>
>     | % Data Checked | TraceIn | Trak | Influence Function | Our Method |
>     |--------------------------|---------|------|---------------------|------------|
>     | 1%                       | 1%      | 12%   | 8%                   | 3.1%       |
>     | 2%                       | 2.5%    | 18.3% | 17%                   | 5%         |
>     | 5%                       | 35.3%   | 23%   | 22.3%                  | 23.6%      |
>     | 10%                      | 42%     | 27%   | 25.3%                  | 38.3%      |
>     | 15%                      | 52.7%   | 39.3% | 38.7%                  | 49.6%      |
>     | 20%                      | 62.3%   | 47.7% | 41.7%                  | 59%        |
>     | 25%                      | 66%     | 53%   | 56%                     | 68%        |
>     | 30%                      | 72%     | 68.7% | 67.7%                  | 75.1%      |
>
> - The above experiments are averaged over 3 random training runs with variance as follows: TraceIn variance = 0.02 , TRAK variance = 0.03, Influence Function variance = 0.02, and f-INE variance =0.01.
> - Additionally, to further scale our method in high-dimensional settings, we have added an additional case study on model's explainability where we use mini-ImageNet dataset (subset of ImageNet Dataset with 100 classes) with Resnet-50 model. The results can be found in Appendix section C of the new version of the PDF.
>
>
> 2. **Separation of the utility and consistency metrics. More unified metric**
> - We thank the reviewer for this suggestion. Note that high utility (recall) and high consistency (low variance) are desired. In our experiments, we show that our method has superior/comparative utility while achieving lower variance.
> - We agree with the reviewer that a unified metric might be useful. We are open to suggestions about designing such a metric from the reviewer and will be happy to include it in future versions.

---

### Official Review · Reviewer_ezeg · 2025-11-06

**Soundness:** 4
**Presentation:** 3
**Contribution:** 3
**Rating:** 8
**Confidence:** 4

**Summary:**

In this paper, estimating the influence of a training data point is reframed as a binary hypothesis test between distributions. The paper shows that instead of treating influence as a deterministic property of a fixed training path, one should take a distributional view that explicitly incorporates training-time randomness (randomness stemming from factors such as optimization, weight initialization, data shuffling, etc, ). They formalize f-inf as the distinguishability between model outputs trained with and without a given data subset. The key idea is that strict total ordering, during ML training for neural nets, is underspecified unless one fixes a testing trade-off. Consequently, they define f-influence using trade-off functions and show that, for a learning procedure that composes, these functions asympotitically tend towards a notion that they term: single scalar Gaussian Influence. The f-INE, is a single-run algorithm that makes this estimate by collecting gradient-similarity statistics with and without a target subset of inputs and then computing test errors across thresholds. Their experiments show that improved stability to randomness compared to baselines and alternatives like TRAK, LESS, etc.

**Strengths:**

- Important and Timely problem: instability of influence scores across random seeds is an important and mostly understudied problem in the literature. This work puts this at center stage and provides an algorithm for attacking this issue.
- Interesting Application of Differential Privacy Ideas: While the connection between influence and differential privacy is not new. The formulation here is quite directly connected and based on ideas from differential privacy. It seems that the key change moving away from a worst-case setting that differential privacy typically requires.
- This approach avoids multi or retraining.
- Compelling LLM Llama-3.1-8B  case study: On Llama-3.1-8B instruction tuning with poisoned, f-INE achieves higher recall of poisoned items and lower coefficient of variation across seeds than LESS (a different approach).

**Weaknesses:**

- This work requires a careful reading of the Dong et. al. paper on f-Differential Privacy. In fact, I would say one main point of this work is to apply the insights from that paper to influence estimation. However, I am not an expert on differential privacy, so I don't know the amount of innovation involved in translating this insight here.

- Decision guidance: Provide a short “operationalization” section: given an estimated $\mu$ how should data curators set Type-I/II budgets to choose deletions under randomness?

Minor issues:
- Need a gap between last sentence of first page and the first sentence of the page 2. Put a space between lines 067 and 068.

**Questions:**

- Decision guidance: Provide a short “operationalization” section: given an estimated $\mu$ how should data curators set Type-I/II budgets to choose deletions under randomness?
- How sensitive are your results to batch size and the projection dimension (for LoRA gradients), and the number/placement of checkpoints? For typical setting, how would you suggest one set these?
- It is somewhat unclear to me whether the negative influence values have a different behavior and subsets with positive influence. I assume because these are distributional tests that there is no difference in implementation of the algorithm between these two groups.

---

> ### Author Response · Authors · 2025-11-20
> **Rebuttal by Authors**
>
> We thank the reviewer for the positive comments about the paper. Specific doubts are clarified below:
>
> 1. **Given an estimated $\mu$ how should data curators set Type-I/II budgets to choose deletions under randomness?**
> - Note that the $\mu$ value is calculated using Type-I error ($\alpha$) and Type-II error ($\beta$) using the equation $\mu = \Phi^{-1}(1-\alpha) - \Phi^{-1}(\beta)$. Now, for a given $\mu$ value, there can be multiple $\alpha$ and $\beta$ values that can result in the same $\mu$ values. Now, a preferable budget for Type-I/II errors depends on the downstream application in a particular context - if coverage is more important, then we would pick a low type II error, or if precision is important, we would pick a low type I error.
> - There might be some confusion from our side in understanding this question. Thus, we request the reviewer to clarify further if the above clarification doesn't answer the question. If it does, then we will add the suggested "operationalization" section in the subsequent version.
>
>
> 2. **Sensitivity of projected gradient dimensions and number of checkpoints**
> - We have added these ablation results in Figure-9 and Figure-10 of Section 4.4 in the new version of the PDF. We request the reviewer to look at the updated version of the PDF.
> - We observe that as the projection dimension $d$ decreases, our method is quite robust with only a slight degradation in performance. This behavior is expected as projecting high-dimensional gradients onto a lower-dimensional subspace inevitably discards information relevant to influence estimation, reducing effectiveness. A similar degradation trend is also observed for the LESS method.
> - Further, we notice that using a higher number of checkpoints gives better utility for f-INE. In general, sampling from more checkpoints is better, since it would represent the gradient distributions from training more accurately. As discussed in Sections 4.4 and 4.5, f-INE's superior utility arises from its ability to pick up on the long tails of gradient distributions; hence, if we sample from less number of checkpoints, we may risk missing out on the long tail signals. Thus, with a low number of checkpoints, f-INE and LESS are quite similar, and f-INE gets increasingly better as we increase the number of checkpoints.
>
>
> 3. **Negative influence values have a different behavior and subsets with positive influence**
> - The reviewer is correct that there is no difference in the implementation of the algorithm between these two groups, as computation is identical.
> - While the computation is identical, their relative significance depends on the application, and typically, variance is more problematic for negative influence data. For example, it is very critical to identify and remove all the negative samples in our task of removing poisoned/mislabeled samples; however, missing some of the positive samples is usually tolerable.

---

### Meta-Review · Area_Chair_kWRg · 2026-01-05

**Summary:**

The reviewers raised the following main concerns, including insufficient empirical evaluations in terms of main experiments and ablation studies, limited discussion of related work, and issues with clarity and paper presentation. In addition, reviewer ap96 raised concerns regarding the paper's central premise.

**Reviewer Concerns:**

The rebuttal addressed the majority of the reviewers' concerns by providing additional experiments and ablations, adding additional discussion on related work, and responding to clarification questions. The rebuttal also address reviewer ap96's concern by clarifying the intended purpose of influence estimation.

**Reviewer Scores:**

If reviewers had been able to participate fully in the discussion, I believe reviewer ap96 might have increased their score from 2 to 4 or 6, while other reviewers' scores would likely remain unchanged.

---

### Decision · Program_Chairs · 2026-01-26

Accept (Poster)